# Mutation analysis in individual circulating tumor cells depicts intratumor heterogeneity in melanoma

Mark Sementsov [1,2], Leonie Ott [1], Julian Kött [2,3], Alexander Sartori [4], Amelie Lusque[5], Sarah Degenhardt [6], Bertille Segier [5], Isabel Heidrich[2,3], Beate Volkmer [6], Rüdiger Greinert[6], Peter Mohr[7], Ronald Simon[8], Julia-Christina Stadler[2,3], Darryl Irwin [9], Claudia Koch[1], Antje Andreas[1], Benjamin Deitert [1], Verena Thewes[10], Andreas Trumpp [10], Andreas Schneeweiss[11], Yassine Belloum[1], Sven Peine[12], Harriett Wikman [1], Sabine Riethdorf [1], Stefan W Schneider [3], Christoffer Gebhardt [2,3,14✉], Klaus Pantel [1,14✉] & Laura Keller [1,2,13,14✉]

## Abstract

**Circulating tumor DNA (ctDNA) is the cornerstone of liquid biopsy diagnostics, revealing clinically relevant genomic aberrations from blood of cancer patients. Genomic analysis of single circulating tumor cells (CTCs) could provide additional insights into intra-patient heterogeneity, but it requires whole-genome amplification (WGA) of DNA, which might introduce bias. Here, we describe a novel approach based on mass spectrometry for mutation detection from individual CTCs not requiring WGA and complex bioinformatics pipelines. After establishment of our protocol on tumor cell line-derived single cells, it was validated on CTCs of 33 metastatic melanoma patients and the mutations were compared to those obtained from tumor tissue and ctDNA. Although concordance with tumor tissue was superior for ctDNA over CTC analysis, a larger number of mutations were found within CTCs compared to ctDNA ($p = 0.039$), including mutations in melanoma driver genes, or those associated with resistance to therapy or metastasis. Thus, our results demonstrate proof-of-principle data that CTC analysis can provide clinically relevant genomic information that is not redundant to tumor tissue or ctDNA analysis.**

**Keywords** Circulating Tumor Cells; Tumor; Heterogeneity; Melanoma
**Subject Categories** Cancer; Methods & Resources

## Introduction

With minimally invasive access to tumor cells or tumor-derived material, liquid biopsy allows a dense monitoring of tumor evolution through the course of treatment (Heidrich et al, 2023; Alix-Panabières and Pantel, 2021; Keller and Pantel, 2019). Mutational profiling has been preferentially performed on ctDNA, the tumor-originating fraction of circulating cell-free DNA (cfDNA), mainly released as small fragments from apoptotic cells into the blood circulation. The growing popularity of ctDNA analysis can be explained by the considerable technological progress made in lowering mutation detection thresholds with highly sensitive methods like digital PCR or adapted next-generation sequencing (NGS) protocols as well as simplified pre-analytical workflows requiring only blood centrifugation and DNA extraction. ctDNA studies have shown a good concordance in the detection of the main driver mutation with tumor tissue in a large, diverse set of tumor entities (Keller et al, 2021; Odegaard et al, 2018). Interestingly, somatic mutations that had been missed in the corresponding tissue sample were also reported, suggesting that mutations in ctDNA can arise from different tumor regions (Heitzer et al, 2019; Parikh et al, 2019; Pereira et al, 2021; Wong et al, 2017; Murtaza et al, 2015). Thereby, liquid biopsy blood analyses have been shown to give a more complete overview of the tumor genome than a single tumor biopsy (Keller and Pantel, 2019; Heidrich et al, 2023; Alix-Panabières and Pantel, 2021).

In addition to ctDNA, CTCs offer the opportunity to assess the genome and transcriptome of intact viable tumor cells with a high metastatic potential and hence to retrieve relevant functional information of living cells. However, individual CTC analysis requires a labor-intensive workflow and in particular genome

[1]Department of Tumor Biology, University Medical Center Hamburg-Eppendorf, Hamburg, Germany. [2]Fleur Hiege Center for Skin Cancer Research, University Medical Center Hamburg-Eppendorf, Hamburg, Germany. [3]Department of Dermatology and Venereology, University Medical Center Hamburg-Eppendorf, Hamburg, Germany. [4]Agena Bioscience GmbH, Hamburg, Germany. [5]Biostatistics & Health Data Science Unit, Institut Claudius-Regaud, IUCT-Oncopole, Toulouse, France. [6]Department of Molecular Cell Biology, Skin Cancer Center Buxtehude, Elbe Kliniken Buxtehude, Buxtehude, Germany. [7]Department of Dermatology, Elbe Kliniken Buxtehude, 21614 Buxtehude, Germany. [8]Institute of Pathology, University Medical Center Hamburg-Eppendorf, Hamburg, Germany. [9]Agena Bioscience, Bowen Hills, QLD, Australia. [10]Division of Stem Cells and Cancer, German Cancer Research Center (DKFZ) and DKFZ-ZMBH Alliance, Heidelberg, Germany. [11]National Center for Tumor Diseases, Heidelberg University Hospital and German Cancer Research Center, Heidelberg, Germany. [12]Department of Transfusion Medicine, University Medical Center Hamburg-Eppendorf, Hamburg, Germany. [13]CRCT, Université de Toulouse, Inserm, CNRS, Université Toulouse III-Paul Sabatier, Centre de Recherches en Cancérologie de Toulouse, Toulouse, France. [14]These authors contributed equally: Christoffer Gebhardt, Klaus Pantel, Laura Keller. ✉E-mail: ch.gebhardt@uke.de; pantel@uke.de; keller.laura@iuct-oncopole.fr

analysis requires a whole-genome amplification (WGA) step which is considered mandatory to get sufficient DNA for downstream molecular analysis like single nucleotide variant (SNV) (Paoletti et al, 2018; Rossi et al, 2022; Rothé et al, 2022) or CNA profiling (Carter et al, 2017; Asante et al, 2023; Paoletti et al, 2018; Fernandez-Garcia et al, 2022; Oulhen et al, 2021). Besides increasing the turnaround time of the analysis, this step is prone to amplification bias, polymerase errors, and allele drop-outs (Lohr et al, 2014; Pailler et al, 2019; Babayan et al, 2016; Cani et al, 2022). Overcoming this technical hurdle would therefore simplify mutation detection in single CTCs, thereby expanding the possibilities to provide in real time a more comprehensive picture of tumor evolution and heterogeneity from liquid biopsy analysis.

Here, we present a simplified, WGA-free workflow for CTC analysis at the single cell level to interrogate multiple particular hotspot mutations. Our method was validated by single cell analysis of 132 CTCs isolated from the peripheral blood of 33 metastatic melanoma patients, and the results were compared to the mutations on ctDNA and tumor tissue, which demonstrated that genomic CTC analysis provides additional clinically relevant genomic information.

# Results

## Mutation detection rate at the single cell level with the modified UltraSEEK® Melanoma protocol

Our modified protocol (see Methods) was first tested on genomic DNA (gDNA) extracted from 5 tumor cell lines (A2058, SKMEL28, SKMEL30, SKMEL2, WM1366) harboring 5 different mutations comprised among the predefined UltraSEEK® Melanoma panel (see Dataset EV 1 or 2 for the list of mutations). We serially diluted gDNA down to the level of a single cell (about 6.6 pg for a diploid cell line) and assessed the protocol in duplicate at each dilution. At 6.6 pg, at least one duplicate was positive according to our mutation calling process based on 'Normalized Intensity' and SNR values (Fig. EV1B) indicating thereby the capacity of our protocol to detect mutations from minute amounts of DNA. Note that at this stage, the 'Normalized Intensity' threshold is defined on water samples (Dataset EV 1).

After setting up our protocol on gDNA, we tested it on single cells not chemically preserved (i.e., alive), referred to hereafter as 'non-fixed' single cells. The median detection rate of mutation (calculated among lysed cells) was 80% and varies from 60% (MAP2K1 P124S_f2) to 100% (NRAS Q61L_r1) (Fig. EV1A,B).

## Mutation detection after CTC enrichment methods with the modified UltraSEEK® protocol

CTC enrichment methods are generally expected to process chemically preserved blood. Chemical preservation can induce diverse DNA damages (Hykin et al, 2015; Quach et al, 2004), potentially compromising the subsequent detection of mutations. As representative for size- and deformability-based CTC enrichment methods, we selected the Parsortix® enrichment device with blood chemically preserved in Transfix® tubes as previous results had demonstrated superior CTC recovery rates versus EDTA-blood

(Koch et al, 2020). Thus, we assessed our protocol on tumor cells from melanoma cell lines spiked into blood from healthy donors, chemically preserved in Transfix® tubes and processed with the Parsortix® CTC enrichment method, mimicking thereby the complete patient workflow (Fig. 1A).

We defined a new normalized intensity threshold based on leukocytes processed through the same workflow (Dataset EV 2). In order to validate the mutation calling process (based on the normalized intensity and the SNR value), we defined a validation set composed of 15 single cells from 2 different cell lines (A2058, SKMEL30) and 12 leukocytes from one healthy donor. We observed 11 positive calls in 7 different mutations assays (CTNNB1_S45P-f1; KIT_V559A-f1; KIT_V559A-f2; KIT_L576P-f2; MAP2K1_P124S-f2; SDHD 'mut1'-f1; SDHD 'mut2'-f1, Dataset EV 3) among the leukocytes. In melanoma tumor cell lines, we did not detect any other mutation apart from the expected BRAF V600E, MAP2K1 P124S (_f1), and NRAS Q61K mutations (Dataset EV 3). While we have not observed any unexpected mutation in 90.8% (69/76) of the assays, 6.6% (5/76) of the assays (CTNNB1_S45P-f1; KIT_L576P-f2; MAP2K1_P124S-f2; SDHD_mut1-f1; SDHD_mut2-f1), present a rate of unexpected mutation of 3.7% (1/27) and 2.63% (2/76) of the assays (KIT_V559A-f1; KIT_V559A-f2) present a rate of unexpected mutation of 11.1% (3/27). Therefore, the mutation calling strategy is validated for the vast majority of the UltraSEEK® Melanoma Panel assays and the assays with positive values on leukocytes were excluded from further analysis.

Of note, when re-analyzing the mutation recovery rates for "non-fixed" single cells with the allele frequency threshold set on leukocytes, the same recovery performance was obtained as with the threshold defined on water controls.

On 'Parsortix' cells, the lysis efficiency (i.e., the number of cells efficiently lysed and with DNA successfully amplified over the total number of cells successfully picked; see Methods for determination of cell lysis status and successful gDNA targeted amplification) was 80.8% ± 10% (mean ± SD). The median detection rate of mutations was 81.5% and varies from 41.1% (MAP2K1 P124S_f1) to 87.5% (NRAS Q61L_r1, NRAS Q61R_r1). Mutation detection rate for each mutation is detailed in Fig. 1A and characteristics of each mutation call (i.e., normalized intensity and SNR) are given in Fig. EV1B. Therefore, the modified UltraSEEK® protocol does not seem to be drastically impacted by chemical preservation of the blood samples.

Overall, these results validate the capability of our protocol to detect different mutations in single CTCs isolated with the Parsortix® CTC enrichment method. We thus extended our study to a cohort of advanced (metastatic, unresectable) melanoma patients and compared the mutations found in CTCs and ctDNA with the same panel.

## CTC detection in the metastatic melanoma cohort

In total, 33 metastatic, non-resectable melanoma patients were enrolled and their clinical characteristics are presented in Table 1. The vast majority of these patients (72.7%) were receiving a first line of therapy (93.7% immunotherapy and 6.3% targeted therapy). BRAF V600E and NRAS mutations (NRAS Q61R, NRAS Q61K) were found in tumor tissue in 35.5% (11/31) and 19.4% (6/31) patients, respectively. Twenty-five patients were sampled before the start of a new line of therapy (three of them were also sampled

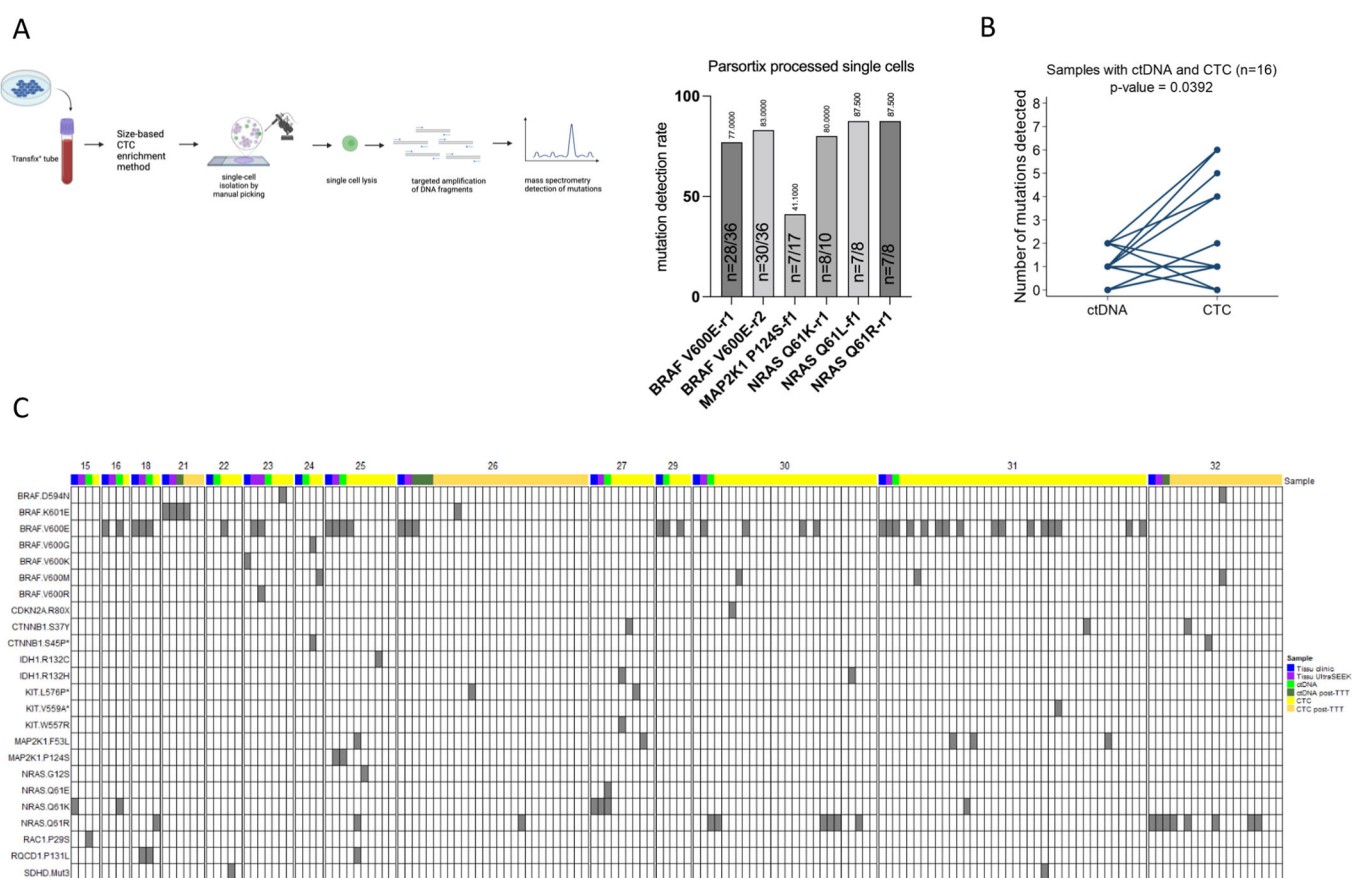

**Figure 1. Melanoma CTC analysis complements ctDNA analysis for mutation detection**

(A) Left: Schematic figure of the Parsortix® enrichment workflow with melanoma tumor cell lines. Left panel: melanoma tumor cells were spiked into 'Transfix' blood, captured in the Parsortix® cassette and harvested on a slide for staining to be picked by manual micromanipulation. Created using BioRender.com. Right panel: Recovery rate obtained for each assay on single cell with Parsortix® enrichment. Detection rate is displayed above the bar and the number of cells with successful mutation detection over the number of efficiently lysed cells is mentioned within bars. Note that recovery data are presented for each PCR assay in case multiple PCR assays for the same mutation are present in the panel. (B) Number of mutations in ctDNA or CTCs in samples with both sufficient amount of cfDNA and at least one CTC efficiently lysed, as defined by internal process controls. Of note, the assays with positive signals observed in the leukocytes validation set have been removed. *P*-value was calculated using the Wilcoxon signed-rank test for paired data. (C) Landscape of the mutations detected in the different melanoma samples (tissue, cfDNA, CTC). On the top of the panel, patient ID is indicated with the type of sample analyzed. We distinguished samples obtained before initiation of treatment from those obtained during treatment. For patient 26, we grouped the 3 samples obtained during treatment. 'Tissue UltraSEEK' stands for the analysis of gDNA from a tumor tissue sample with UltraSEEK® Melanoma Panel. The assays with positive values on leukocytes validation set are marked with an asterisk. SDHD.Mut3 stands for SDHD mutation Chr11:111, 957,544 C > T. Source data are available online for this figure.

during the treatment) and eight only during the treatment, totaling 39 blood samples. Melanoma CTCs were captured with the Parsortix® device (whose CTC recovery performance was $72 \pm 12\%$ (Fig. EV2A,B). CTCs were detected in 69.2% (27/39 (95%CI: 52.4–83.0) samples and the median CTC number per sample was 4 (range: 1–55), reaching a total of 250 CTCs.

With a median clinical follow-up of 11 months [95%CI: 9.1; 19.7], the number of CTCs detected per patient on the blood sample before immunotherapy initiation tended to be associated with a worse prognosis (progression-free survival HR (95%CI):1.04 (0.99–1.09); $p = 0.16$ and overall survival HR (95% CI): 1.08 (1.00–1.17); $p = 0.053$ in univariable Cox proportional hazard model). Clinical characteristics of patients included in this study are provided in Table 1. No association between specific metastatic sites with CTC detection was found in this pilot study.

## Mutational analysis of individual CTCs in melanoma patients

For mutational analysis with the modified UltraSEEK® Melanoma Panel, 73.2% (183/250) CTCs were manually picked and 72.1% (132/183) cells were successfully lysed, a rate similar to the one previously observed on melanoma cell lines. In terms of blood samples, 70.4% (19/27) had at least one CTC efficiently lysed.

Seven assays that presented positive signals in leukocytes were excluded from the analysis of mutations in patients and affected 5 CTCs. Overall, we found mutations in 55.6% (15/27) of the samples corresponding to 57/132 (43.2%) of CTCs efficiently lysed. In the vast majority (52/57, 91.2%) of the CTCs, only one mutation was found while 5/57 (8.8%) CTCs had 2 mutations or more. Parameters (normalized intensity and SNR) to call mutations within CTCs are provided in Dataset EV 4.

**Table 1. Patients clinical characteristics.**

| | Total (*N* = 33) | Patients included in survival analysis (*N* = 23) |
|---|---|---|
| Age: median (range) | 60 (41–88) | 61.0 (41.0–88.0) |
| **Gender** | | |
| Female | 12 (36.4%) | 9 (39.1%) |
| Male | 21 (63.6%) | 14 (60.9%) |
| **Primary melanoma site** | | |
| CUP | 4 (13.3%) | 4 (18.2%) |
| Cutaneous | 20 (66.7%) | 12 (54.5%) |
| Mucosal | 3 (10.0%) | 3 (13.6%) |
| Uveal | 3 (10.0%) | 3 (13.6%) |
| ND | 3 | 1 |
| **AJCC classification (at diagnosis)** | | |
| IIC | 1 (3.0%) | 0 |
| III | 4 (12.1%) | 4 (17.4%) |
| IV | 28 (84.8%) | 19 (82.6%) |
| **Metastatic classification (at diagnosis)** | | |
| M0 | 7 (21.2%) | 6 (26.1%) |
| M1a | 2 (6.1%) | 2 (8.7%) |
| M1b | 4 (12.1%) | 3 (13.0%) |
| M1c | 13 (39.4%) | 7 (30.4%) |
| M1d | 7 (21.2%) | 5 (21.7%) |
| **Prior therapies** | | |
| No | 24 (72.7%) | 17 (73.9%) |
| Yes | 9 (27.3%) | 6 (26.1%) |
| **Mutation status (tissue)** | | |
| BRAF.K601E | 1 (3.2%) | 0 |
| BRAF.V600E | 11 (35.5%) | 6 (28.6%) |
| BRAF.V600K | 2 (6.5%) | 2 (9.5%) |
| NRAS.Q61K | 3 (9.7%) | 3 (14.3%) |
| NRAS.Q61R | 3 (9.7%) | 2 (9.5%) |
| WT (BRAF, NRAS) | 11 (35.5%) | 8 (38.1%) |
| ND | 2 | 2 |
| **Baseline therapy** | | |
| Immunotherapy | 30 (93.7%) | 23 (100.0%) |
| Targeted therapy | 2 (6.3%) | 0 |
| ND | 1 | 0 |
| **Timing of blood samples (*n* = 33)** | | |
| Baseline only | 22 (66.7%) | 23 (100.0%) |
| Baseline + Post-TT | 3 (9.1%) | 0 |
| Post-TT only | 8 (24.2%) | 0 |
| **Time from tissue analysis to blood sampling[a] (*n* = 25)** | | |
| 0–2 months | 11 (47.8%) | 10 (47.6%) |
| >2 months | 12 (52.2%) | 11 (52.4%) |
| Missing | 2 | 2 |

*AJCC* American Joint Committee on Cancer, *CUP* Cancer of Unknown Primary.
[a]For blood samples taken before treatment initiation.

## Comparative mutational analysis of CTCs and ctDNA in melanoma patients

In parallel to CTCs, the same mutations were screened in cfDNA using the standard UltraSEEK® Melanoma Panel (Dataset EV 5). The description of mutation detection performance within each liquid biopsy (LB) analyte is recapitulated in Table 2. In total, 59.0% (23/39) samples had both CTCs detected and a sufficient amount of cfDNA, four samples had only CTCs, nine only cfDNA and three neither. No correlation was found between the number of CTCs and cfDNA concentration (Spearman coefficient: 0.20; $p = 0.24$). Despite a higher potential of cfDNA to provide mutational information (82.1% (32/39) of samples had a sufficient amount of cfDNA versus 48.7% (19/39) with at least one CTC picked and efficiently lysed), the percentage of samples with at least one mutation detected was similar between CTC and cfDNA (38.5% (15/39) versus 48.7% (19/39)) and the number of different mutations was higher in CTCs (Tables 2 and 3). Notably, among the 16 samples that presented both a sufficient amount of cfDNA and at least one CTC picked and efficiently lysed, a significantly larger number of mutations per sample was also found within CTCs (range: 0–6) in comparison to ctDNA (range: 0–2; $p = 0.0392$) (Fig. 1B).

Among those 16 samples (Fig. 1C), 2 (12.5%) were fully concordant (i.e., all mutations detected are identical), 4 (25%) partially concordant (i.e., at least one mutation in common) and 10 (62.5%) were discordant (i.e., no mutation in common). In those latter 10 samples, 5 had no mutation in ctDNA while at least one mutation was detected in CTC, 3 had a mutation (RAC1 P29S, NRAS Q61E, and MAP2K1 P124S) detected in ctDNA not found in CTC and 2 had different mutations between CTCs and ctDNA. Thus, CTC analysis provides additional information to ctDNA in 11/16 (68.8%) of the samples. The mutations that could be only found in CTCs in some of these 11 samples were canonical mutations in *BRAF* (V600E and K601E) or *NRAS* (Q61R and Q61K) genes (7 samples), less frequent mutations in *BRAF* (V600G, V600M, D594N) or *NRAS* (G12S) genes (5 samples), activating mutations in *CTNNB1* gene (2 samples) and the MAP2K1 F53L activating mutation (3 samples) (Fig. 1C).

Overall, these results demonstrate that CTC analysis could bring non-redundant information to ctDNA on melanoma driver genes or those associated with resistance to therapy or metastasis.

## Concordance level between liquid biopsy analytes and tumor tissue

We assessed the concordance level for *NRAS* and *BRAF* mutations between each LB analyte and the tumor tissue since these two genes are commonly screened in clinical routine. Twenty patients with a sufficient amount of cfDNA and 11 patients with at least one CTC picked and lysed in the blood sample before therapy initiation were analyzed for tissue concordance, respectively. Median time from tissue analysis to blood sampling was 6.2 months (range: 0; 41) for those samples. Concordance was defined as follows: the *BRAF* or *NRAS* mutation was commonly shared with the tissue and each LB analyte or both tissue and LB analyte were negative for mutations in these two genes. 85% (17/20, 95%CI: 62.1–96.8) of the blood samples were concordant between ctDNA and tumor tissue. CTC

**Table 2. Comparison between ctDNA and CTC analysis for mutation detection.**

| | ctDNA analysis (N = 39) | CTC analysis (N = 39) |
|---|---|---|
| Number of samples with informative LB analyte[a] | 32/39 (82.1%, 95%CI: 66.5–92.5) | 19/39 (48.7%, 95%CI: 32.4–65.2) |
| Number of samples with at least one mutation detected | 19/39 (48.7%, 95%CI: 32.4–65.2) | 15/39 (38.5%, 95%CI: 23.4–55.4) |
| Number of samples where two mutations or more were detected | 8/39 (20.5%, 95%CI: 9.3–36.5) | 8/39 (20.5%, 95%CI: 9.3–36.5) |
| Median number of mutations (when detected) per sample | 1 (range: 1–2) | 2 (range: 1–6) |
| Total number of different mutations | 8 | 17 |

[a]"Informative" LB analyte means at least one CTC picked and efficiently lysed or presence of sufficient amount of cfDNA.

**Table 3. Mutations found in ctDNA and CTCs.**

| | Number of ctDNA samples (N = 32[a]) | Number of CTC samples (N = 19[b]) |
|---|---|---|
| Any mutation | 19 (59.4%) | 15 (78.9%) |
| BRAF.D594N | – | 2 (10.5%) |
| BRAF.K601E | 2 (6.3%) | 2 (10.5%) |
| BRAF.V600E | 11 (34.4%) | 6 (31.6%) |
| BRAF.V600G | – | 1 (5.3%) |
| BRAF.V600M | – | 4 (21.1%) |
| CDKN2A.R80X | – | 1 (5.3%) |
| CTNNB1.S37Y | – | 3 (15.8%) |
| IDH1.R132C | – | 1 (5.3%) |
| IDH1.R132H | – | 2 (10.5%) |
| KIT.W557R | – | 1 (5.3%) |
| MAP2K1.P124S | 1 (3.1%) | – |
| MAP2K1.F53L | – | 3 (15.8%) |
| NRAS.G12S | – | 1 (5.3%) |
| NRAS.G13D | – | 1 (5.3%) |
| NRAS.Q61E | 1 (3.1%) | – |
| NRAS.Q61K | 3 (9.4%) | 2 (10.5%) |
| NRAS.Q61R | 3 (9.4%) | 5 (26.3%) |
| RAC1.P29S | 5 (15.6%) | – |
| RQCD1.P131L | 1 (3.1%) | 1 (5.3%) |
| SDHD. Chr11:111, 957,544 C > T | – | 2 (10.5%) |

[a]In samples with sufficient amount of cfDNA (32 samples).
[b]In samples with at least one CTC picked and efficiently lysed (19 samples).

concordance with tissue (i.e., at least one CTC concordant) was observed in 45.5% (5/11, 95%CI: 16.7–76.6) of the samples.

Other mutations than the one identified in the tissue were found in 9/20 samples with ctDNA and in 8/11 samples with CTC. To check if these additional mutations found in each LB analyte were or were not present in a tumor tissue, we re-analyzed the genomic DNA from the available tumor tissues with the UltraSEEK® Melanoma Panel. In 3/7 samples (two tissue samples were missing for this analysis), the additional mutations found in ctDNA were also present in the tumor tissue (Appendix Fig. S1, Table EV 1 and Dataset EV 5). Except for one BRAF V600E mutation that could be also found in the tissue of one patient, the mutations found in CTCs were not found in the tumor tissue. These "private" CTC mutations affected significantly mutated genes in melanoma

(BRAF, NRAS, MAP2K1, KIT) or were in genes associated with a more aggressive phenotype like CTNNB1. Thus, ctDNA and CTC analyses of blood from melanoma patients can reveal mutations not found by bulk analysis of randomly selected primary tumor tissue of the same patient.

One clinical case illustrates how liquid biopsy and CTC analysis in particular could provide more information about intra-patient heterogeneity and about the metastatic clone relevant for disease progression. Patient MEL30 was diagnosed with a metastatic cutaneous melanoma wild-type (WT) for BRAF and NRAS genes in its primary tumor but a BRAF V600E mutation was present in a resected lymph node metastasis. This clinically relevant mutation characterizing the metastasis was also detectable in CTCs but not in ctDNA from the same sample, supporting the view that CTCs might represent enriched selective information not present on "bulk" ctDNA. In addition, an NRAS Q61R mutation not present in the tumor tissue was found in both ctDNA and CTCs (Fig. 1C).

## Application of the modified UltraSEEK® protocol to other cancer entities and other CTC enrichment methods

As a proof-of-concept, we tested whether our protocol might be applicable to other cancer entities with the use of another UltraSEEK® mutation panel. After testing mutation detection rate of 4 mutations of the UltraSEEK® Lung Panel on gDNA and non-fixed single cells (Fig. EV3A,C), we tested the detection of EGFR T790M and L858R mutations on single tumor cells from the H1975 cell line processed through the Parsortix® enrichment method and obtained between 57% (_f2) and 78.5% (_f1) detection rate for EGFR T790M mutation and 64.3% for EGFR L858R mutation (Fig. EV3B,C). Of note, no other positive signal was observed on the entire panel for 4 tumor cells and 5 leukocytes (Dataset EV 7).

Finally, we sought to explore whether the modified UltraSEEK® protocol could also be applicable with CTCs isolated with another platform like CellSearch®. The BRAF V600E mutation was successfully detected in 88.9% of SKMEL28 tumor cells tested with the UltraSEEK® Melanoma Panel (Fig. 2A). We then analyzed CTCs from two patients (MEL21 and MEL31). In patient MEL21, we managed to find in one individual CTC the BRAF K601E corresponding to the tumor tissue (Figs. 2B and EV4A). Interestingly, we concomitantly detected another NRAS Q61R mutation in this cell. In CTCs from patient MEL31, with a BRAF V600E-mutated tumor, the BRAF V600E mutation was detected in the majority of the CTC analyzed (5/7) and another MAP2K1 N382H mutation was also detected in 2/7 CTCs (Figs. 2C and EV4B). As the CellSearch® enrichment platform has been extensively used to detect CTCs in breast cancer, we also tested whether we could detect mutations in genes of therapeutic relevance in this setting. After verifying that we could detect PI3KCA E545K

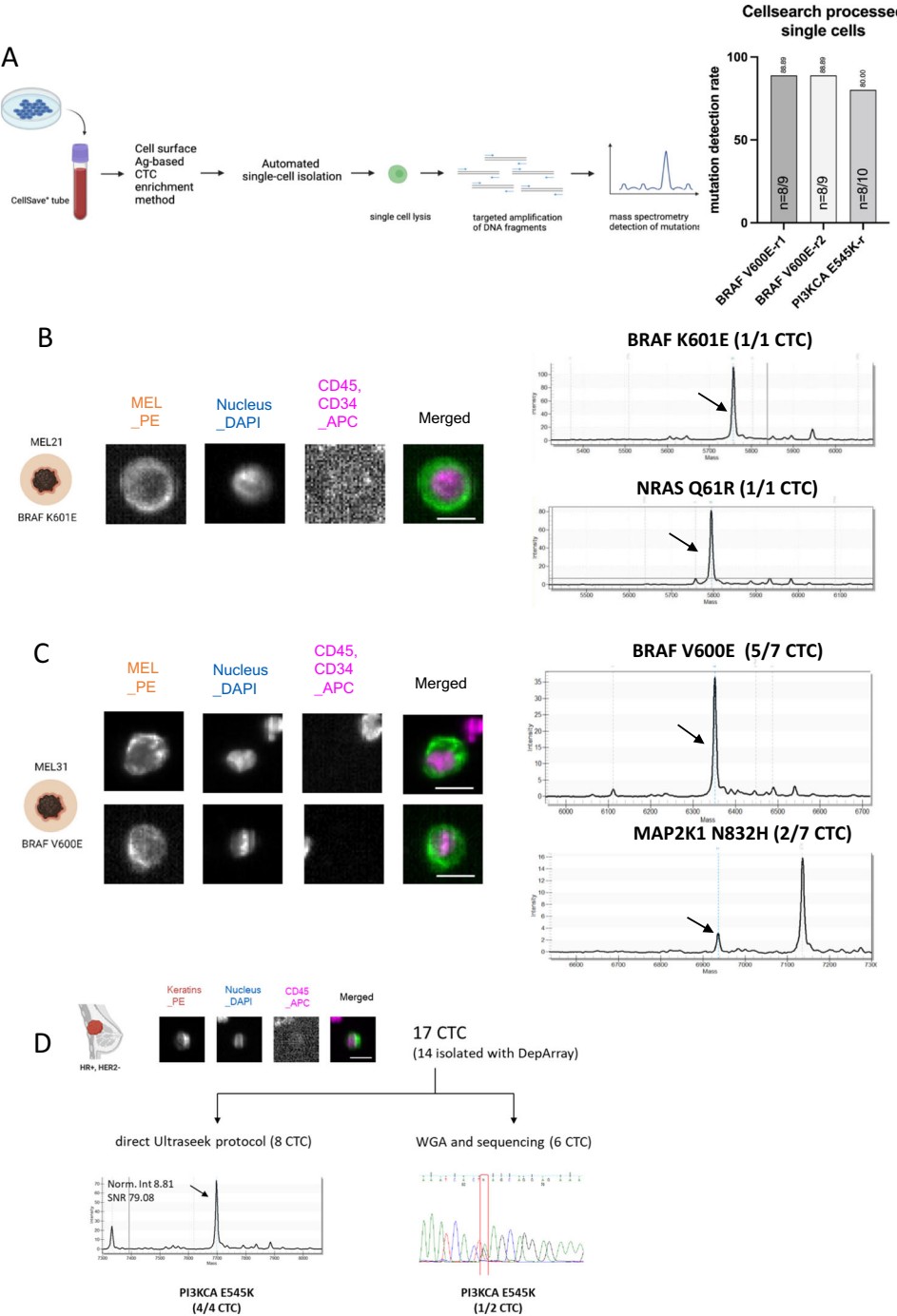

**Figure 2. Mutation detection on different individual CTCs isolated with CellSearch® and DEPArray™ combined workflow.**

(**A**) Left: Schematic figure of the CellSearch®/DEPArray™ enrichment workflow with melanoma and breast tumor cell lines. Left panel: melanoma or breast tumor cells were spiked into 'CellSave' blood, captured on the CellSearch® platform and isolated as single cells with the DEPArray™ system. Created using BioRender.com. Right panel: Recovery rate obtained for each assay on single cell with CellSearch®/DEPArray™ enrichment. The detection rate is displayed above the bar and the number of cells with successful mutation detection over the number of efficiently lysed cells is mentioned within bars. Note that recovery data are presented for each PCR assay in case multiple PCR assays for the same mutation are present in the panel. (**B, C**) Illustrative picture of melanoma CTCs and of the mutation calls in 2 melanoma patients (B. patient MEL21; C. Patient MEL31). Scale bar represents *ca.* 10 μm. (**D**) Illustrative picture of breast cancer CTCs and its subsequent analysis with UltraSEEK® protocol (for 8 CTCs) or sequencing after WGA (for 6 CTCs). Successful cell lysis and gDNA amplification was obtained in 4/8 CTCs with UltraSEEK® protocol and from 2/6 CTCs after WGA quality control. PI3KCA E545K mutation was detected in 4/4 CTC and 1/2 CTC, respectively. Scale bar represents *ca.* 10 μm. Source data are available online for this figure.

mutation from MCF7 cells alive or processed with the CellSearch®/DEPArray™ workflow (70% and 80% detection rate, respectively, Figs. 2A and EV3A), we assessed CTCs obtained from a breast cancer patient (primary tumor: ER+, PR+, HER2 1+, Ki67 15%) with the UltraSEEK® Lung Panel, as this panel interrogates several genes of therapeutic interest like ERBB2, PIK3CA in breast cancer. With the modified UltraSEEK® protocol, we detected a PIK3CA E545K mutation in 4 CTCs (Figs. 2D and EV4C). Importantly, this mutation was also separately detected in one CTC out of two by sequencing its WGA product, confirming our results (Fig. 2D).

## Discussion

Mutation detection among individual CTCs is challenging in melanoma and other solid tumors. After CTC isolation, standard workflow typically requires WGA followed by NGS analysis and usually takes about 5 to 10 days depending on the protocols used. Here, we report on the successful development of a simplified workflow that amplifies the regions of interest, allowing the interrogation of multiple hotspot mutations at the single cell level without requiring prior WGA, does not depend on NGS and that can be performed in 3 days, therefore presenting a substantial gain in the turn-around time of the analysis (Fig. EV5). To do so, we selected a cost-effective solution initially developed to interrogate multiple hotspot mutations in ctDNA with high sensitivity (Lamy et al, 2020) and reproducibility (Weber et al, 2020), independently validated in metastatic melanoma (Gray et al, 2019) or lung cancer patients (Lamy et al, 2020; Belloum et al, 2020).

We obtained efficient rates of single-cell lysis and DNA amplification (superior to 70% in CTCs from patients) and detection rates superior to 70% for several mutations on individual tumor cells. This protocol was mainly assessed with CTCs isolated by the Parsortix® device, one of the two FDA-cleared CTC enrichment methods. Preliminary results obtained on two tumor cell lines and CTCs isolated from three patients with the CellSearch®/DEPArray™ workflow (Fig. 2) suggest that the modified UltraSEEK® protocol might as well be applicable on multiple mutations with this other FDA-cleared CTC-enrichment platform. In the two melanoma cases where CTCs were both isolated with CellSearch® or Parsortix® enrichment methods, the driver mutation present in the tumor tissue was detected on CTCs enriched with both methods. Nevertheless, we also report mutations exclusively found in the 'CellSearch-CTC' that were not found in the 'Parsortix-CTC'. This can be explained by the fact that both methods capture (different) CTCs based on different enrichment principles (membrane antigen for CellSearch® and size/deform-ability for Parsortix®), which might explain the heterogeneity of results. Finally, our results also suggest that this protocol could be applied to other solid tumor entities like lung or breast cancers.

With this new protocol, we could study multiple hotspot mutations in 13 genes from 132 individual melanoma CTCs of 33 metastatic melanoma patients. To our best knowledge, only BRAF V600E had so far been tested within melanoma CTCs (Sakaizawa et al, 2012). Although timing between blood sampling and tissue analysis should be considered, the concordance level of 45% between CTCs and tumor tissue analysis suggests a higher degree of intra-patient heterogeneity in melanoma than the one observed with bulk sequencing of tumor tissues (Chang et al, 2020). The

genomic information of CTCs may therefore not only originate from the tumor tissue that was analyzed from the patient but could also be released by small metastatic subclones or occult metastatic lesions, as it was already demonstrated by autopsy studies for ctDNA in melanoma (Wong et al, 2017) and several other clinical entities (Parikh et al, 2019; Pereira et al, 2021; Murtaza et al, 2015). CTC analysis could therefore provide a more comprehensive picture of intra-patient heterogeneity.

Mutations found in ctDNA and CTCs and concordance with tumor tissue have yet been rarely compared in the same cancer patients. First, the concordance level between ctDNA and the tumor tissue was consistent with previous reports (Diefenbach et al, 2020; Calapre et al, 2019; Santiago-Walker et al, 2016) but superior to the one between CTCs and the tumor tissue. This difference could be explained by the fact that ctDNA analysis corresponds to a kind of bulk sequencing approach with a sensitivity limit that depends on the assay employed (down to 0.1% of variant allele frequency with UltraSEEK® assay for certain mutations) and reflects a larger pool of tumor cells than a single CTC. A variant allele frequency of 0.1% observed in ctDNA is estimated to correspond to a tumor volume of $10 \, cm^3$ (i.e., 300 million malignant cells) in NSCLC (Abbosh et al, 2023, 2017) although this also remains to be verified in melanoma (Braune et al, 2020) since shedding capacity can also vary between tumor type and localization. For CTCs, the stochasticity related to the low volume of blood sampled in case of a low number of cells together with the difference in sensitivity level between a single cell analysis and a bulk sequencing approach is to be considered for the interpretation of concordance with the tumor tissue. Second, our results also demonstrate that CTC analysis brings complementary information to ctDNA in almost 70% of the cases with some CTC 'private' mutations in genes implicated in immune evasion or tumor dissemination mechanisms like CTNNB1 (Karachaliou et al, 2022; Damsky et al, 2011; Spranger et al, 2015; Massi et al, 2017; Lin et al, 2020) and CDKN2A (Zeng et al, 2018). These findings appear particularly relevant in this context since CTCs are intact viable cells that reflect an active process of tumor dissemination and a more aggressive phenotype, as corroborated by their almost significant association with worse overall survival in this cohort and by other previous reports including our own in melanoma (Gorges et al, 2019; Lucci et al, 2020; Gray et al, 2015; Khoja et al, 2012). CTCs might also originate from a different metastatic subclone than the one(s) shedding ctDNA. In some patients, we could also exclusively detect mutations in melanoma driver genes (BRAF, NRAS, MAP2K1, KIT) suggesting that CTC analysis could complement ctDNA analysis for relevant clinical implications in personalized medicine in oncology.

Our study presents some limitations, the first one being the small size of our patient cohort. Follow-up samples would have also been interesting to capture tumor evolution under therapy and larger cohorts of patients from different clinical entities should now be studied in order to better understand the complementarity of CTC and ctDNA mutational profiling. Second, mutation detection from single CTCs remains technically challenging at different steps. Single-cell isolation is the first bottleneck that needs to be further improved by more efficient automated methods like the DEPArray™ used in our study for single cell isolation of tumor cell-line cells and CTCs isolated by the CellSearch® system. Even if we managed to pick 73.2% of the potential CTCs we detected, the workflow used for the melanoma cohort relies on manual picking whose success rate can be dependent on user's expertise. Lysis efficiency is the next step

that can lead to some loss even if we managed to lyse more than 70% of patient CTCs and finally, the mutation detection efficiency of our protocol is around 81.5% (among lysed cells) based on the results of our tumor cell lines experiments. Therefore, the combination of these different technical steps impact our mutation detection efficiency from CTC. Nevertheless, the observed detection rate of mutations in CTCs of this cohort (43%, 57/132) could be as well explained by the limited number of mutations interrogated by the panel (albeit designed to detect the most frequent ones) and biological factors such as the unknown percentage of cells within each tumor bearing the mutation and the fact that CTCs are tumor cells selected through the tumor dissemination process. Consequently, all these parameters play a role in the appraisal of intra-tumor heterogeneity and in the comparison of the molecular content of CTCs with tumor tissue and/or ctDNA. Specificity of our mutation calling approach also plays a role in that matter. Our mutation calling strategy is based on two parameters (normalized intensity and SNR) that reflect two complementary characteristics of the peak. While the vast majority (90.8%) of the assays did not present any signal in leukocytes used as negative control, we observed positive signals in some leukocytes for 7 assays that we excluded from CTC analysis.

Solutions to interrogate a larger number of mutations or other kind of genetic events are therefore needed for a more comprehensive exploration of the genome content of CTCs. In this line, attempts to adapt next-generation sequencing (NGS) methods to single CTCs without WGA have been described. To our best knowledge, the latter have so far only been successful on pools of CTCs from scarce cases of pancreatic (Yu et al, 2020) or colorectal (Liebs et al, 2021) cancers. High throughput sequencing methods applicable to single cells and using targeted amplification of genomic regions of interest are thus promising. But, they are not yet adapted to the low number of CTCs present in a usual blood draw because of the high level of cell drop-outs which necessitates a large input of single cells (Morita et al, 2020; Gao et al, 2019).

Overall, we developed an original simplified workflow without the need of WGA and complex bioinformatics to interrogate multiple clinically relevant mutations in individual CTCs that demonstrate the utility of CTC analysis to describe intra-tumor heterogeneity more comprehensively in melanoma patients. We foresee that the mutational profiling of CTCs could complement ctDNA genotyping in order to more easily decipher the clonal architecture and to identify the most aggressive clones, CTC analysis also bringing the advantage to correlate the phenotype to a particular genotype with multi-omic approaches. In addition, single-cell CTC mutational analysis could circumvent the challenges associated with clonal hematopoiesis for interpretation of ctDNA results. Early stages and detection of minimal residual disease would be particularly interesting to study, which is limited however by the extremely low CTC counts in non-metastatic cancer patients. However, molecular characterization of CTCs is limited by the CTC capture rate, independent from the downstream molecular assay used for genomic profiling. Technologies to capture higher CTC numbers and to improve the sorting of single CTCs for downstream analyses are currently being established (Keller and Pantel, 2019) which, in combination with the presented platform for genomic single cell analysis may open new avenues for future investigation and potentially envision clinical applications.

# Methods

## Patient cohort

This study included 21 metastatic, non-resectable melanoma patients from the University Skin Cancer Center, University Medical Center Hamburg-Eppendorf, Hamburg, Germany. All patients provided informed consent for participating in this study, which was approved by the ethical commissions of the Hamburger Ärztekammer (General Medical Council Hamburg) (PV5392). This study also included 12 metastatic melanoma patients from the Department of Dermatology, Elbe Kliniken Buxtehude, Buxtehude, Germany, who also provided informed consent to participate in this study. Inclusion criteria were: (i) a confirmed diagnosis of melanoma according to the AJCC version 8 melanoma staging and classification (ii) at least 18 years of age. All histologic types of melanoma, including mucosal and uveal melanoma were eligible for inclusion. Exclusion criteria were the presence of an autoimmune disease, HIV, hepatitis B or C, pregnancy, or concomitant systemic therapy for melanoma.

The blood sample from the metastatic breast cancer patient for CTC detection with the CellSearch® system was obtained from the Gynecologic Oncology, National Center for Tumor Diseases, University of Heidelberg.

The experiments performed on human-derived blood samples conformed to the principles set out in the WMA Declaration of Helsinki and the Department of Health and Human Services Belmont Report.

## Tumor cell lines and culture

The melanoma-derived cell lines SKMEL28 (carrying the BRAF V600E mutation) and WM1366 (NRAS Q61L), the breast cancer cell lines (MCF7 (PIK3CA E545K) and T47D (PIK3CA H1047R)) and the non-small cell lung cancer cell line (H1975 (EGFR T790M, EGFR L858R) were purchased from the ATCC (ATCC, Manassas, VA, USA). The melanoma-derived cell lines SKMEL2 (NRAS Q61R), SKMEL30 (NRAS Q61K) that were chosen for analysis were a kind gift from Dr. Alex Bauer and Prof. Stefan W. Schneider. The melanoma-derived cell line A2058 (BRAF V600E, MAP2K1 P124S) was kindly provided by Dr. Thomas Schlange within the Cancer ID consortium.

Cell lines were tested for mycoplasma contamination and recently authenticated. Cells were cultured in cell culture flasks under standard conditions in humidified incubators at 37 $\circ$C with 10% $CO_2$ or 5% $CO_2$ depending on the medium. RPMI and DMEM media (Gibco-Life Technologies, Darmstadt, Germany) were employed as recommended by ATCC and fortified with 10% fetal bovine serum (Gibco-Life Technologies, Darmstadt, Germany, 1% L-glutamine (Gibco-Life Technologies, Darmstadt, Germany) and 1% penicillin/streptomycin (Gibco-Life Technologies, Darmstadt, Germany). Cell passaging was performed at 70% confluency.

gDNA isolation from tumor cell lines was performed using the NucleoSpin tissue kit (Macherey Nagel, Düren, Germany) according to manufacturer's instructions.

## Spiking of healthy donor blood with tumor cell line cells

Cell line cells were prepared for spiking experiments by washing with 2 x PBS (Gibco-Life Technologies, Darmstadt, Germany) and incubating with 1X Citrate buffer (diluted from 10X Citrate buffer: 1.35 M KCL,

150 mM Na-Citrate into PBS) for 5 min at room temperature prior to resuspending in culture medium. The cell suspension was centrifuged at $190 \times g$ for 5 min after which the supernatant was discarded, and the cells were re-suspended in fresh culture medium.

For the evaluation of cell recovery after Parsortix® enrichment, a defined number of cells (30) were spread to a petri dish filled with medium, manually counted and picked under a light microscope. Defined cell counts were added to Transfix® chemically preserved blood samples from healthy donors (HD). For the cell lines that were only used to evaluate our mutation detection protocol, a higher non-defined number of cells was directly spiked into the blood of healthy donors. Following blood spiking, the blood samples were processed through Parsortix® enrichment (see next sections for details).

For the evaluation of our mutation detection protocol on CellSearch®, samples from healthy donors spiked with cells from tumor-cell lines were drawn into CellSave™ (Menarini Silicon Biosystems) preservation tube to be processed on the CellSearch® (Menarini Silicon Biosystems) system.

## CTC enrichment methods

The Parsortix® system (ANGLE plc, Guildford, UK) is a benchtop microfluidic device designed for the size and deformability-based capture of CTC from whole blood (Hvichia et al, 2016). It processes the blood through an enclosed disposable cassette (6.5 μm gap) with a controlled liquid flow rate defined by a pre-set separation pressure. Based on previous results (Koch et al, 2020), we selected the separation rate 99 mbar to process Transfix® blood samples. After enrichment within the cassette, the liquid flow is reversed and the captured tumor cells are flushed out. Cells were directly harvested into cytospin funnels, centrifuged onto a glass slide ($190 \times g$, 3 min), dried overnight, and stored at $-80\,°C$ until further processing (see next section for details).

The CellSearch Melanoma cell kit (Menarini Silicon Biosystems) were used to enrich CTCs from blood samples spiked with melanoma cells (SKMEL28) or from the blood of two melanoma patients (Riethdorf et al, 2007). Image gallery analyses were performed by experienced scientist (Sabine Riethdorf). Melanoma CTCs were defined as NG2-positive, CD45 and CD34-negative cells with a nuclear DAPI staining.

## Melanoma CTC immunofluorescence identification assay

Tumor cells isolated with the Parsortix® system were identified by immunofluorescence. Dried cytospin slides stored at $-80\,°C$ were brought to room temperature (RT) and fixed with 0.5% PFA (SigmaAldrich, Steinheim, Germany) for 10 min. The samples were washed three times with 0.5 mL of 1x-PBS prior to application of 10% AB-serum/PBS (BioRad, Rüdigheim, Germany) for blocking (20 min at RT). The following antibodies were incubated for 60 min at room temperature: MCAM-AF488 (1/100, clone P1H12, Biolegend, San Diego, CA, USA), NG2-AF488 (1/50, clone 9.2.27, BD, Franklin Lakes, NJ, USA), S100B-DyLight550 (1/100, clone 4C4.9 + S100B/1012, Novus Biologicals, Wiesbaden-Nordenstadt, Germany), Melanoma-marker cocktail-DyLight550 (MART-1, Tyrosinase, gp100 Antibodies 1/100 (clones A103 + T311 + HMB45) Novus Biologicals, Wiesbaden-Nordenstadt, Germany), CD45-PerCP (1/200, clone HI30, Biolegend, San Diego, CA, USA), CD16-PerCP (1/200, clone 3G8, Biolegend, San Diego, CA, USA),

together with DRAQ5 (1/5000 Cell Signaling Technology, Danvers, MA, USA). After 3 washes with 0.5 mL of 1x-PBS, cytospins were covered with Prolong Gold Antifade Reagent (Thermo Fisher Scientific, Dreieich, Germany), sealed with a coverslip and examined by fluorescence microscopy (Axio Observer, Zeiss).

MCAM/NG2-positive and/or Melanoma-marker/S100B-positive, DRAQ5-positive, CD45-negative cells with intact morphology were defined as tumor cells.

## Single-cell micromanipulation

After screening of Parsortix® slides, tumor cells or identified CTC were manually picked from the glass slide by a micromanipulator (Eppendorf). Single cells were transferred to the cap of a 200 μl-PCR tube under visual control by a nuclear fluorescent DRAQ5 staining to ensure that whole nuclei were captured.

For single cells isolated from tumor cell lines that were not processed through CTC enrichment methods, a cell suspension was spread into a petri dish filled with PBS for these cells to be manually picked under a light microscope. Single cells were transferred to a cap of a 200 μl-PCR tube under visual control in bright field to ensure that the cell was captured.

## DEPArray™ isolation

CellSearch® cartridges were stored at $4\,°C$ in a dark environment before being processed (within the next 30 days after CellSearch® enrichment) to recover single CTCs with the DEPArray™ system (Menarini Silicon Biosystems) according to the manufacturer's instructions. Cartridge content was transferred into the DEPArray™ cartridge, which facilitates the selection of individual cells based on immunofluorescent staining criteria and cell morphology. After imaging, individual selected cells were routed for single-cell isolation and recovery. After isolation, single cells were stored at $-20\,°C$ or immediately lysed.

## Single-cell lysis

For cell lysis, 3 μl of lysis buffer (Appendix Table S1) were applied to the single cell for 5 min on ice immediately after its transfer into the cap of a 200 μl-PCR tube. After a centrifugation step ($10,000 \times g$, 10 min, $4\,°C$), the lysis solution was incubated for 10 h at $42\,°C$, followed by inactivation at $65\,°C$ for 30 min and denaturation at $95\,°C$ for 15 min. Cell lysates are then stored at $-80\,°C$ until further analysis.

## UltraSEEK® Melanoma and Lung Panels on single cells

The UltraSEEK® Melanoma and Lung Panels (Agena Bioscience, San Diego, CA, USA) are designed to detect multiple hotspot mutations of melanoma or lung cancer entities from ctDNA, respectively (Dataset EV 2 and EV 6 for the complete list of mutations). We modified the wet-lab workflow for UltraSEEK® panels for the use on single cells. Therefore, instead of one, we performed two successive multiplex PCR reactions to generate a sufficient amount of PCR products. Reaction mixes and PCR cycling conditions are provided in Appendix Tables S2. For variant detection the remainder default workflow was performed as previously described (Mosko et al, 2016) consisting of

depurination, single-base extension with biotinylated chain terminator oligonucleotides specific to the mutant allele and capture by streptavidin-coated magnetic beads. Liquid handling of the captured extended oligonucleotides, transfer onto a SpectroCHIP® Arrays and data acquisition by mass-spectrometry was automatically performed with the MassARRAY® System (Agena Bioscience, San Diego, USA) according to manufacturer's instructions. Data analysis was performed using Somatic Variant Caller (SVR) software version 1.0.5 (Agena Bioscience, San Diego, USA).

Successful cellular lysis and amplification of gDNA was indicated by the area value of process controls (internal assays specific to beta-actin housekeeping gene). The threshold of the area was calculated from the according values of lysis buffer samples (NTC) and corresponds to the mean+10 SD (Dataset EV 2, 3, 6 and 7).

For successfully lysed and DNA-amplified cells, mutation calling integrates two parameters related to the intensity (i.e., signal-to-noise ratio (SNR) and normalized intensity ('allele frequency') from the SVR output) of the peak observed at the expected molecular mass for the mutant allele. SNR, which represents the ratio of the peak intensity of the mutant allele to the intensity of the surrounding background, was required to be above 10. Normalized intensity of mutant allele peak was calculated from peak height values of the target signal and of 5 capture controls (biotin-labeled, non-reactive oligonucleotides added to the extension reaction to control streptavidin-bead capture and elution). For the recovery experiments performed with cell lines processed through CTC enrichment methods and for the patient CTCs, the threshold of the normalized intensity was calculated from the according values of leukocytes (41 and 19 leukocytes for the melanoma and the lung UltraSEEK® Melanoma and Lung Panels, respectively) and corresponds to the mean+10 SD (Dataset EV 2 and 6).

Experiments on single tumor cells and leukocytes were not blinded to the researchers.

## Plasma analysis and cfDNA extraction

Blood was collected into 7.5 ml EDTA tubes (S-Monovette® 7.5 ml Sarstedt, Germany) and centrifuged twice (10 min at $300 \times g$, followed by 10 min at $1800 \times g$) to isolate plasma within 2 h after the blood draw. Plasma was stored at $-80\,°C$ until extraction. CfDNA was extracted from 4 ml of plasma using the QIAamp Circulating Nucleic Acid Kit (Qiagen, Hilden, Germany) according to manufacturer's instructions and stored at $-80\,°C$ until quantification and quality assessment. CfDNA was quantified using Qubit Fluorometer 2.0 (Thermo Fisher). The quality (assessment of the cfDNA amount coming from the white blood cell contamination) was assessed by the LiquidIQ® Panel (Agena Bioscience, San Diego, CA, USA) according to manufacturer's instructions (Lamy et al, 2020). The optimal amount of cfDNA input used in the UltraSEEK® Melanoma assay was optimized according to the Qubit fluorometer concentration.

## ctDNA analysis with UltraSEEK® Melanoma assay

The UltraSEEK® Melanoma assay for ctDNA was already described (Gray et al, 2019). 12–15 ng of cfDNA was used to perform the PCR reaction according to manufacturer's instructions (Agena Bioscience, San Diego, CA, USA). Briefly, 12 to 15 ng of cfDNA were incubated in a Labcycler (Sensoquest, Germany) initially at 95 °C for 2 min, followed by forty-five cycles of PCR (95 °C for 30 s,

56 °C for 30 s, and 72 °C for 1 min). The initial PCR was finalized by incubation of 5 min at 72 °C. The resulting products were subjected to shrimp alkaline phosphatase treatment for 40 min at 37 °C and to denaturation for 5 min at 85 °C. Next, single-base extension with biotinylated chain terminator nucleotides specific to the mutant allele was performed at 95 °C for 30 s, followed by 40 cycles at 94 °C for 5 s with five nested cycles (of 52 °C for 5 s and 80 °C for 5 s) and a final incubation at 72 °C for 3 min. The single-base extended oligonucleotides were captured by streptavidin-coated magnetic beads and subsequently pelleted using a magnet and re-suspended in 13 μl of biotin competition solution, which was then incubated at 90 °C for 5 min. Finally, the extension products were transferred into the MassARRAY® System for automated sample handling including desalting with anion exchange resin, dispensing analytes onto the SpectroCHIP® Array (Agena Bioscience, San Diego, CA, USA) and data acquisition via MALDI-TOF mass spectrometry, as described by Mosko et al, 2016. Data analysis was performed using MassARRAY® Typer Analyzer software version 5.0 (Agena Bioscience) as well as the automated Somatic Variant Report (SVR) software version 1.2 (Agena Bioscience) with plugin for the UltraSEEK® Melanoma Panel v2.0.

## Tissue analysis with Agena protocol

Tumorous regions on the histopathological slides were identified by an experienced scientist (Ronald Simon, UKE). Tumor DNA was isolated with the QIAamp DNA FFPE Tissue Kit. 15 ng of tumor DNA per sample were treated identically to the ctDNA samples (see above).

## Statistical analysis

Data was described using usual descriptive statistics. Continuous variables were summarized by median and range (min-max) and qualitative variables by frequencies and percentages. Comparisons of continuous variables between groups were performed using Kruskal-Wallis test and correlation between continuous variables was measured using Spearman coefficient. The concordance between tumor tissue and each analyte (CTC or ctDNA from blood sample before treatment initiation) were evaluated by patient with the rate of concordance and its 95% confidence interval (CI) on the basis of an exact binomial distribution. The concordance between CTCs and ctDNA was evaluated by blood sample and comparison between the number of mutations detected between both was performed using the Wilcoxon signed-rank test for paired data. Overall survival (OS) and progression-free survival (PFS) were calculated from immunotherapy initiation and estimated by the Kaplan–Meier method in patients treated with immunotherapy and with a sample available before treatment initiation. The associations between continuous variables and survival endpoints (OS and PFS) were assessed using the Cox proportional hazard model.

All statistical tests were two-sided, and a $p$ value < 0.05 was considered to be statistically significant. Statistical analyses were carried out using Stata Statistical Software version 16 (College Station, TX, StataCorp LLC).

## Whole genome amplification of breast cancer CTCs

Breast cancer CTCs were isolated using the DEPArray™ system as described above. Six single CTCs were used for WGA, using the

## The paper explained

### Problem

Metastatic spread of tumor cells through the bloodstream is the main cause of cancer-related death. Needle biopsies of metastatic tumor lesions are invasive procedures with potentially harmful side effects for the patient such as bleeding or infections, and some locations are difficult to reach. Thus, « liquid biopsies » on blood samples have been developed as a minimally invasive alternative to obtain comprehensive information on the tumor lesion(s) in an individual patient and refers to the analysis of circulating tumor cells (CTCs) or circulating cell-free DNA fragments released from tumors (ctDNA). Compared to ctDNA, CTCs molecular analysis presents a more complicated workflow.

### Results

Here, we successfully developed a new, simplified method to interrogate clinically actionable alterations in individual circulating tumor cells (CTCs) present in peripheral blood of cancer patients as potential precursors of distant metastases. As proof-of-concept, we analyzed CTCs primarily from metastatic melanoma patients in comparison to circulating cell-free DNA fragments released from tumors (ctDNA) and the corresponding tumor tissue. The results obtained on CTCs provided additional information not obtained by ctDNA and tumor tissue analysis.

### Impact

Our results demonstrate the feasibility of our simplified protocol and uncover a potential added value of CTC analysis in detecting genomic information relevant to the development of cancer metastasis.

## For more information

www.elbs.eu

www.uke.de/english/departments-institutes/institutes/tumor-biology/index.html

www.eumelareg.org.

## Data availability

This study includes no data deposited in external repositories.

The source data of this paper are collected in the following database record: biostudies:S-SCDT-10_1038-S44321-024-00082-6.

## Peer review information

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

Ampli1™ WGA Kit (Menarini Silicon Biosystems), following the manufacturer's instructions. Using the Ampli1™ QC Kit (Menarini Silicon Biosystems), following the manufacturer's instructions, the quality of the WGA product was assessed to decide whether it could be used for sequencing. 2/6 cells showed 4/4 PCR bands, as evaluated on a 2% agarose gel, indicating good DNA quality of the WGA product. Its concentration was determined using the Qubit™ 1X dsDNA High Sensitivity Kit (Thermo Fisher Scientific, Dreieich, Germany).

## Targeted sequencing of breast cancer CTC derived WGA product

20 ng of the WGA product of the 2 breast cancer CTCs showing high DNA quality were used for preamplification of the region of interest by PCR with the following primers: Fw: GGGAAAATGACAAAGAA-CAGCTC; Rev: CATTTTAGCACTTACCTGTGAC. The PCR product was separated on a 2% agarose gel and bands of appropriate size were cut out. Isolation of the DNA from the agarose gel was performed using the NucleoSpin Gel and PCR Clean-up Kit (Macherey-Nagel, Düren, Germany), following the manufacturer's instructions. After determination of the DNA concentration using the Qubit™ 1X dsDNA High Sensitivity Kit, 50 ng of the PCR product were used to prepare the Sanger sequencing, using the BigDye™ Terminator v1.1 Cycle Sequencing Kit (Thermo Fisher Scientific, Dreieich, Germany). Following the sequencing reaction, the DNA was precipitated by the addition of 2.5-fold volume of abs. ethanol and 0.1-fold volume of 3 M sodium acetate. After centrifugation at $25,000 \times g$ for 30 min at RT, the supernatant was discarded. The DNA pellets were washed with 125 µl of 70% ethanol and centrifuged again at $25,000 \times g$ for 20 min at RT. The supernatant was removed and the pellets air-dried at RT. The dried pellets were subjected to Sanger sequencing after re-constitution.

tumor cells identifies distinct copy-number profiles in patients with chemosensitive and chemorefractory small-cell lung cancer. Nat Med 23:114–119

Chang GA, Wiggins JM, Corless BC, Syeda MM, Tadepalli JS, Blake S, Fleming N, Darvishian F, Pavlick A, Berman R et al (2020) TERT, BRAF, and NRAS mutational heterogeneity between paired primary and metastatic melanoma tumors. J Invest Dermatol 140:1609–1618.e7

Damsky WE, Curley DP, Santhanakrishnan M, Rosenbaum LE, Platt JT, Gould Rothberg BE, Taketo MM, Dankort D, Rimm DL, McMahon M et al (2011) β-catenin signaling controls metastasis in Braf-activated Pten-deficient melanomas. Cancer Cell 20:741–754

Diefenbach RJ, Lee JH, Menzies AM, Carlino MS, Long GV, Saw RPM, Howle JR, Spillane AJ, Scolyer RA, Kefford RF et al (2020) Design and testing of a custom melanoma next generation sequencing panel for analysis of circulating tumor DNA. Cancers 12:2228

Fernandez-Garcia D, Nteliopoulos G, Hastings RK, Rushton A, Page K, Allsopp RC, Ambasager B, Gleason K, Guttery DS, Ali S et al (2022) Shallow WGS of individual CTCs identifies actionable targets for informing treatment decisions in metastatic breast cancer. Br J Cancer 127:1858–1864

Gao Y, Maria A, Na N, da Cruz Paula A, Gorelick AN, Hechtman JF, Carson J, Lefkowitz RA, Weigelt B, Taylor BS et al (2019) V211D mutation in MEK1 causes resistance to MEK inhibitors in colon cancer. Cancer Discov 9:1182–1191

Gorges K, Wiltfang L, Gorges TM, Sartori A, Hildebrandt L, Keller L, Volkmer B, Peine S, Babayan A, Moll I et al (2019) Intra-patient heterogeneity of circulating tumor cells and circulating tumor DNA in blood of melanoma patients. Cancers 11:E1685

Gray ES, Reid AL, Bowyer S, Calapre L, Siew K, Pearce R, Cowell L, Frank MH, Millward M, Ziman M (2015) Circulating melanoma cell subpopulations: their heterogeneity and differential responses to treatment. J Invest Dermatol 135:2040–2048

Gray ES, Witkowski T, Pereira M, Calapre L, Herron K, Irwin D, Chapman B, Khattak MA, Raleigh J, Hatzimihalis A et al (2019) Genomic analysis of circulating tumor DNA using a melanoma-specific UltraSEEK oncogene panel. J Mol Diagn 21:418–426

Heidrich I, Deitert B, Werner S, Pantel K (2023) Liquid biopsy for monitoring of tumor dormancy and early detection of disease recurrence in solid tumors. Cancer Metastasis Rev 42:161–182

Heitzer E, Haque IS, Roberts CES, Speicher MR (2019) Current and future perspectives of liquid biopsies in genomics-driven oncology. Nat Rev Genet 20:71–88

Hvichia GE, Parveen Z, Wagner C, Janning M, Quidde J, Stein A, Müller V, Loges S, Neves RPL, Stoecklein NH et al (2016) A novel microfluidic platform for size and deformability based separation and the subsequent molecular characterization of viable circulating tumor cells. Int J Cancer 138:2894–2904

Hykin SM, Bi K, McGuire JA (2015) Fixing formalin: a method to recover genomic-scale DNA sequence data from formalin-fixed museum specimens using high-throughput sequencing. PLoS ONE 10:e0141579

Karachaliou GS, Alkallas R, Carroll SB, Caressi C, Zakria D, Patel NM, Trembath DG, Ezzell JA, Pegna GJ, Googe PB et al (2022) The clinical significance of adenomatous polyposis coli (APC) and catenin Beta 1 (CTNNB1) genetic aberrations in patients with melanoma. BMC Cancer 22:38

Keller L, Belloum Y, Wikman H, Pantel K (2021) Clinical relevance of blood-based ctDNA analysis: mutation detection and beyond. Br J Cancer 124:345–358

Keller L, Pantel K (2019) Unravelling tumour heterogeneity by single-cell profiling of circulating tumour cells. Nat Rev Cancer 19:553–567

Khoja L, Lorigan P, Zhou C, Lancashire M, Booth J, Cummings J, Califano R, Clack G, Hughes A, Dive C (2012) Biomarker utility of circulating tumor cells in metastatic cutaneous melanoma. J Invest Dermatol 133:1582–1590

Koch C, Joosse SA, Schneegans S, Wilken OJW, Janning M, Loreth D, Müller V, Prieske K, Banys-Paluchowski M, Horst LJ et al (2020) Pre-analytical and analytical variables of label-independent enrichment and automated detection of circulating tumor cells in cancer patients. Cancers 12:442

Lamy P-J, van der Leest P, Lozano N, Becht C, Duboeuf F, Groen HJM, Hilgers W, Pourel N, Rifaela N, Schuuring E et al (2020) Mass spectrometry as a highly sensitive method for specific circulating tumor DNA analysis in NSCLC: a comparison study. Cancers 12:3002

Liebs S, Eder T, Klauschen F, Schütte M, Yaspo M-L, Keilholz U, Tinhofer I, Kidess-Sigal E, Braunholz D (2021) Applicability of liquid biopsies to represent the mutational profile of tumor tissue from different cancer entities. Oncogene 40:5204–5212

Lin SY, Chang S-C, Lam S, Irene Ramos R, Tran K, Ohe S, Salomon MP, Bhagat AAS, Teck Lim C, Fischer TD et al (2020) Prospective molecular profiling of circulating tumor cells from patients with melanoma receiving combinatorial immunotherapy. Clin Chem 66:169–177

Lohr JG, Adalsteinsson VA, Cibulskis K, Choudhury AD, Rosenberg M, Cruz-Gordillo P, Francis JM, Zhang C-Z, Shalek AK, Satija R et al (2014) Whole-exome sequencing of circulating tumor cells provides a window into metastatic prostate cancer. Nat Biotechnol 32:479–484

Lucci A, Hall CS, Patel SP, Narendran B, Bauldry JB, Royal RE, Karhade M, Upshaw JR, Wargo JA, Glitza IC et al (2020) Circulating tumor cells and early relapse in node-positive melanoma. Clin Cancer Res 26:1886–1895

Massi D, Romano E, Rulli E, Merelli B, Nassini R, De Logu F, Bieche I, Baroni G, Cattaneo L, Xue G et al (2017) Baseline β-catenin, programmed death-ligand 1 expression and tumour-infiltrating lymphocytes predict response and poor prognosis in BRAF inhibitor-treated melanoma patients. Eur J Cancer 78:70–81

Morita K, Wang F, Jahn K, Hu T, Tanaka T, Sasaki Y, Kuipers J, Loghavi S, Wang SA, Yan Y et al (2020) Clonal evolution of acute myeloid leukemia revealed by high-throughput single-cell genomics. Nat Commun 11:5327

Mosko MJ, Nakorchevsky AA, Flores E, Metzler H, Ehrich M, van den Boom DJ, Sherwood JL, Nygren AOH (2016) Ultrasensitive detection of multiplexed somatic mutations using MALDI-TOF mass spectrometry. J Mol Diagn 18:23–31

Murtaza M, Dawson S-J, Pogrebniak K, Rueda OM, Provenzano E, Grant J, Chin S-F, Tsui DWY, Marass F, Gale D et al (2015) Multifocal clonal evolution characterized using circulating tumour DNA in a case of metastatic breast cancer. Nat Commun 6:8760

Odegaard JI, Vincent JJ, Mortimer S, Vowles JV, Ulrich BC, Banks KC, Fairclough SR, Zill OA, Sikora M, Mokhtari R et al (2018) Validation of a plasma-based comprehensive cancer genotyping assay utilizing orthogonal tissue- and plasma-based methodologies. Clin Cancer Res 24:3539–3549

Oulhen M, Pawlikowska P, Tayoun T, Garonzi M, Buson G, Forcato C, Manaresi N, Aberlenc A, Mezquita L, Lecluse Y et al (2021) Circulating tumor cell copy-number heterogeneity in ALK-rearranged non-small-cell lung cancer resistant to ALK inhibitors. NPJ Precis Oncol 5:67

Pailler E, Faugeroux V, Oulhen M, Mezquita L, Laporte M, Honoré A, Lecluse Y, Queffelec P, NgoCamus M, Nicotra C et al (2019) Acquired resistance mutations to ALK inhibitors identified by single circulating tumor cell sequencing in ALK-rearranged non-small-cell lung cancer. Clin Cancer Res 25:6671–6682

Paoletti C, Cani AK, Larios JM, Hovelson DH, Aung K, Darga EP, Cannell EM, Baratta PJ, Liu C-J, Chu D et al (2018) Comprehensive mutation and copy number profiling in archived circulating breast cancer tumor cells documents heterogeneous resistance mechanisms. Cancer Res 78:1110–1122

Parikh AR, Leshchiner I, Elagina L, Goyal L, Levovitz C, Siravegna G, Livitz D, Rhrissorrakrai K, Martin EE, Van Seventer EE et al (2019) Liquid versus tissue biopsy for detecting acquired resistance and tumor heterogeneity in gastrointestinal cancers. Nat Med 25:1415–1421

Pereira B, Chen CT, Goyal L, Walmsley C, Pinto CJ, Baiev I, Allen R, Henderson L, Saha S, Reyes S et al (2021) Cell-free DNA captures tumor heterogeneity and driver alterations in rapid autopsies with pre-treated metastatic cancer. Nat Commun 12:3199

Quach N, Goodman MF, Shibata D (2004) In vitro mutation artifacts after formalin fixation and error prone translesion synthesis during PCR. BMC Clin Pathol 4:1

Riethdorf S, Fritsche H, Müller V, Rau T, Schindlbeck C, Rack B, Janni W, Coith C, Beck K, Jänicke F et al (2007) Detection of circulating tumor cells in peripheral blood of patients with metastatic breast cancer: a validation study of the CellSearch system. Clin Cancer Res 13:920–928

Rossi T, Angeli D, Tebaldi M, Fici P, Rossi E, Rocca A, Palleschi M, Maltoni R, Martinelli G, Fabbri F et al (2022) Dissecting molecular heterogeneity of circulating tumor cells (CTCs) from metastatic breast cancer patients through copy number aberration (CNA) and single nucleotide variant (SNV) single cell analysis. Cancers 14:3925

Rothé F, Venet D, Peeters D, Rouas G, Rediti M, Smeets D, Dupont F, Campbell P, Lambrechts D, Dirix L et al (2022) Interrogating breast cancer heterogeneity using single and pooled circulating tumor cell analysis. NPJ Breast Cancer 8:79

Sakaizawa K, Goto Y, Kiniwa Y, Uchiyama A, Harada K, Shimada S, Saida T, Ferrone S, Takata M, Uhara H et al (2012) Mutation analysis of BRAF and KIT in circulating melanoma cells at the single cell level. Br J Cancer 106:939–946

Santiago-Walker A, Gagnon R, Mazumdar J, Casey M, Long GV, Schadendorf D, Flaherty K, Kefford R, Hauschild A, Hwu P et al (2016) Correlation of BRAF mutation status in circulating-free DNA and tumor and association with clinical outcome across four BRAFi and MEKi clinical trials. Clin Cancer Res 22:567–574

Spranger S, Bao R, Gajewski TF (2015) Melanoma-intrinsic β-catenin signalling prevents anti-tumour immunity. Nature 523:231–235

Weber S, Spiegl B, Perakis SO, Ulz CM, Abuja PM, Kashofer K, van der Leest P, Azpurua MA, Tamminga M, Brudzewsky D et al (2020) Technical evaluation of commercial mutation analysis platforms and reference materials for liquid biopsy profiling. Cancers 12:1588

Wong SQ, Raleigh JM, Callahan J, Vergara IA, Ftouni S, Hatzimihalis A, Colebatch AJ, Li J, Semple T, Doig K et al (2017) Circulating tumor DNA analysis and functional imaging provide complementary approaches for comprehensive disease monitoring in metastatic melanoma. JCO Precis Oncol 1:1–14

Yu J, Gemenetzis G, Kinny-Köster B, Habib JR, Groot VP, Teinor J, Yin L, Pu N, Hasanain A, van Oosten F et al (2020) Pancreatic circulating tumor cell detection by targeted single-cell next-generation sequencing. Cancer Lett 493:245–253

Zeng H, Jorapur A, Shain AH, Lang UE, Torres R, Zhang Y, McNeal AS, Botton T, Lin J, Donne M et al (2018) Bi-allelic loss of CDKN2A initiates melanoma invasion via BRN2 activation. Cancer Cell 34:56–68.e9

## Acknowledgements

We warmly thank all patients and family who have accepted to participate in this study. This project was partially funded by the Hiege Stiftung. LK was financially supported by ITMO Cancer "Plan Cancer 2014–2019" from INSERM and received funding from Fondation de France. LK and KP were financially supported by KMU-innovativ-23 no.031B0843D. SR and KP received funding from the Deutsche Krebshilfe (Förderschwerpunktprogramm 'Translationale Onkologie'; Grant 70114705). KP was financially supported by the ERC Advanced Investigator Grant INJURMET (Nr.834974). We acknowledge financial support from the Open Access Publication Fund of UKE - Universitätsklinikum Hamburg-Eppendorf.

## Author contributions

**Mark Sementsov**: Validation; Investigation; Visualization; Writing—original draft; Writing—review and editing. **Leonie Ott**: Validation; Investigation; Visualization; Writing—original draft; Writing—review and editing. **Julian Kött**: Investigation; Project administration. **Alexander Sartori**: Conceptualization; Resources; Validation; Project administration; Writing—review and editing. **Amelie Lusque**: Formal analysis; Methodology. **Sarah Degenhardt**: Investigation; Project administration. **Bertille Segier**: Formal analysis. **Isabel Heidrich**: Investigation; Project administration. **Beate Volkmer**: Investigation; Project administration. **Rüdiger Greinert**: Investigation; Project administration. **Peter Mohr**: Investigation; Project administration. **Ronald Simon**: Investigation. **Julia-Christina Stadler**: Investigation; Project administration. **Darryl Irwin**: Methodology. **Claudia Koch**: Conceptualization; Investigation; Methodology. **Antje Andreas**: Investigation. **Benjamin Deitert**: Investigation. **Verena Thewes**: Investigation; Project administration. **Andreas Trumpp**: Investigation; Project administration. **Andreas Schneeweiss**: Investigation; Project administration. **Yassine Belloum**: Investigation. **Sven Peine**: Investigation. **Harriett Wikman**: Investigation. **Sabine Riethdorf**: Investigation; Methodology. **Stefan W Schneider**: Investigation; Project administration. **Christoffer Gebhardt**: Resources; Supervision; Funding acquisition. **Klaus Pantel**: Conceptualization; Resources; Supervision; Funding acquisition; Writing—review and editing. **Laura Keller**: Conceptualization; Supervision; Writing—original draft; Writing—review and editing.

Source data underlying figure panels in this paper may have individual authorship assigned. Where available, figure panel/source data authorship is listed in the following database record: biostudies:S-SCDT-10_1038-S44321-024-00082-6.

## Funding

## Disclosure and competing interests statement

MS, LO, JK, AL, BS, IH, JCS, AA, YB, HW, BD, SR, SD, BV, GR, RS, SP, KP, and LK declare no competing interest. AS and DI are employees of Agena Bioscience and hold shares of Mesa Labs. PM declares participation in Data Safety Monitoring Advisory Boards for MSD, Pierre Fabre, GSK, Roche, Bristol Myers Squibb, Novartis, Sanofi, Beiersdorf, Almirall, Hermal, AMGEN, and Sun-Pharma. CG is a member of the advisory board of, and has received honoraria and travel expenses from Almirall, Amgen, Beiersdorf, BioNTech, Bristol-Myers Squibb, Immunocore, Janssen, MSD Sharp & Dohme, Novartis, Pierre-Fabre, Roche, Sanofi Genzyme, SUN Pharma, Sysmex, ouside the submitted work; CG holds shares of Dermagnostix.

# Expanded View Figures

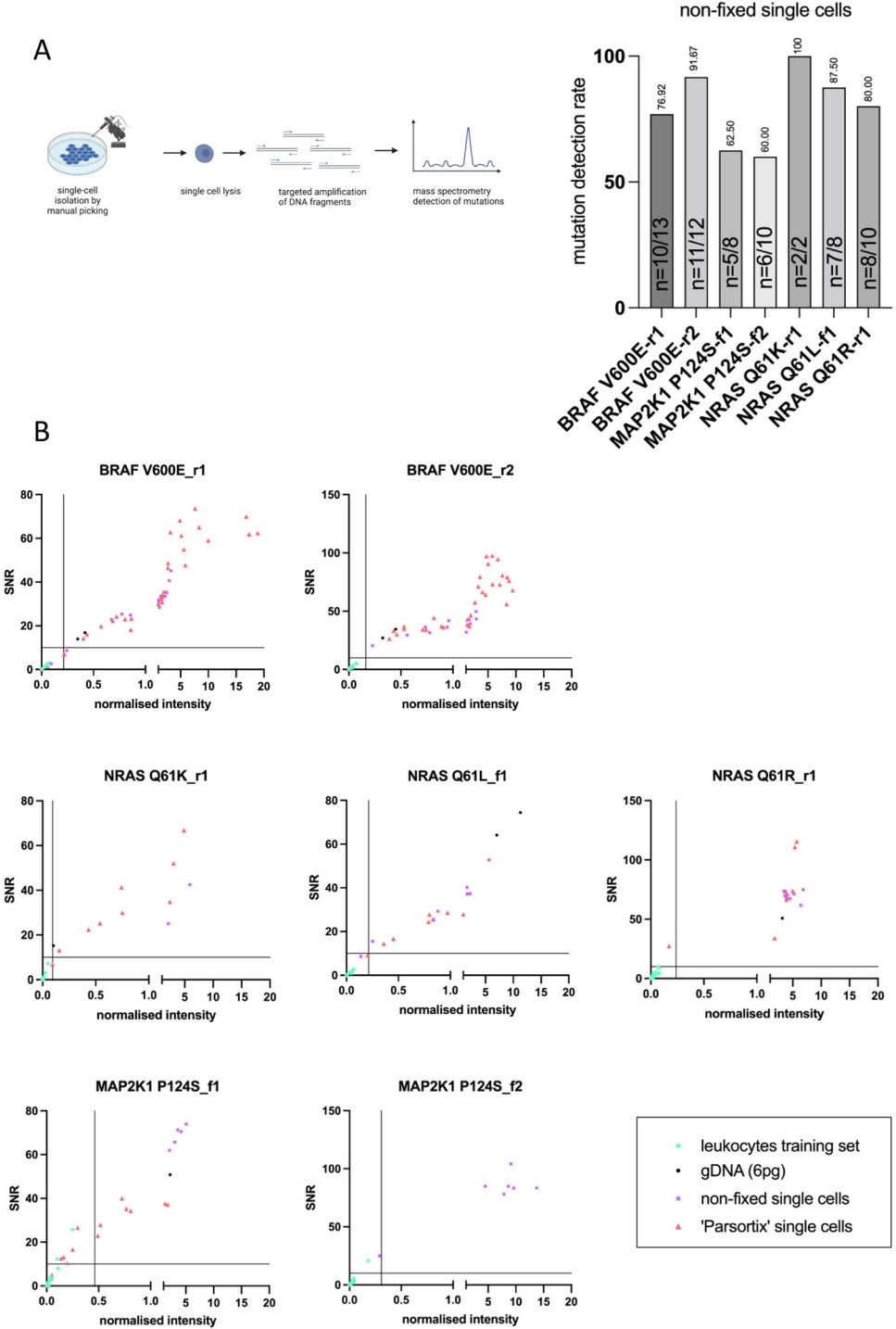

**Figure EV1.  Mutation detection at the single cell level with the UltraSEEK Melanoma Panel.**

(**A**) The UltraSEEK® Melanoma Panel was tested on 5 mutations (certain mutations are detected by 2 different assays in the same panel) with our modified protocol on individual cells from melanoma tumor cell lines not chemically preserved. The number of cells with successful mutation detection over the number of efficiently lysed cells is mentioned. Created with BioRender.com. (**B**) Normalized intensity and signal to noise ratio (SNR) values obtained for each PCR assay for gDNA tested at 6.6 pg input in duplicate, not chemically preserved single cells and 'Parsortix' processed single cells. The horizontal line represents the threshold value of SNR (set to 10) and of the normalized intensity calculated on leukocytes. For MAP2K1 P124S_f2, only non-fixed single cells data are displayed due to the positive signal observed on one leukocyte that led to excluding this assay from further analysis. Source data are available online for this figure.

A

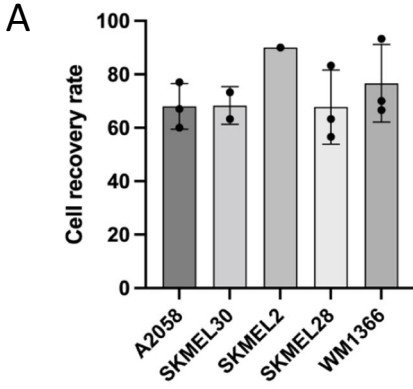

B

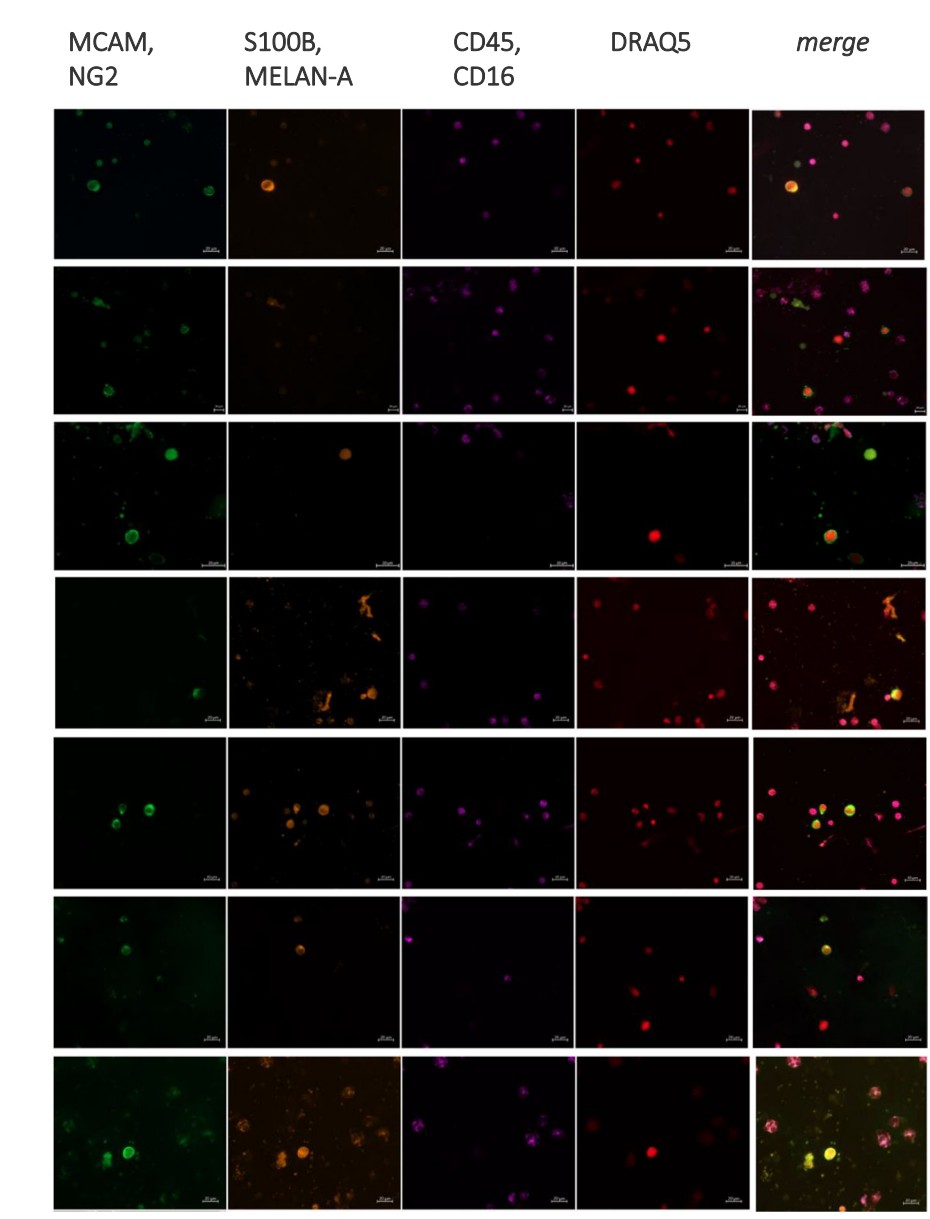

**Figure EV2.  Melanoma CTC enrichment with Parsortix® workflow.**

(**A**) Cell recovery rate for different melanoma cell lines. Recovery rate was calculated from spiking 30 melanoma cells into blood of healthy donors. Each dot represents one technical replicate, bar height represents the mean of recovery rate and error bars Standard Deviation. (**B**) Illustration of melanoma CTCs found in patients. CTCs were defined as nucleated elements (DRAQ5 positive) either MCAM/NG2-positive and/or Melanoma-marker/S100B-positive but CD45/CD16-negative cells. Source data are available online for this figure.

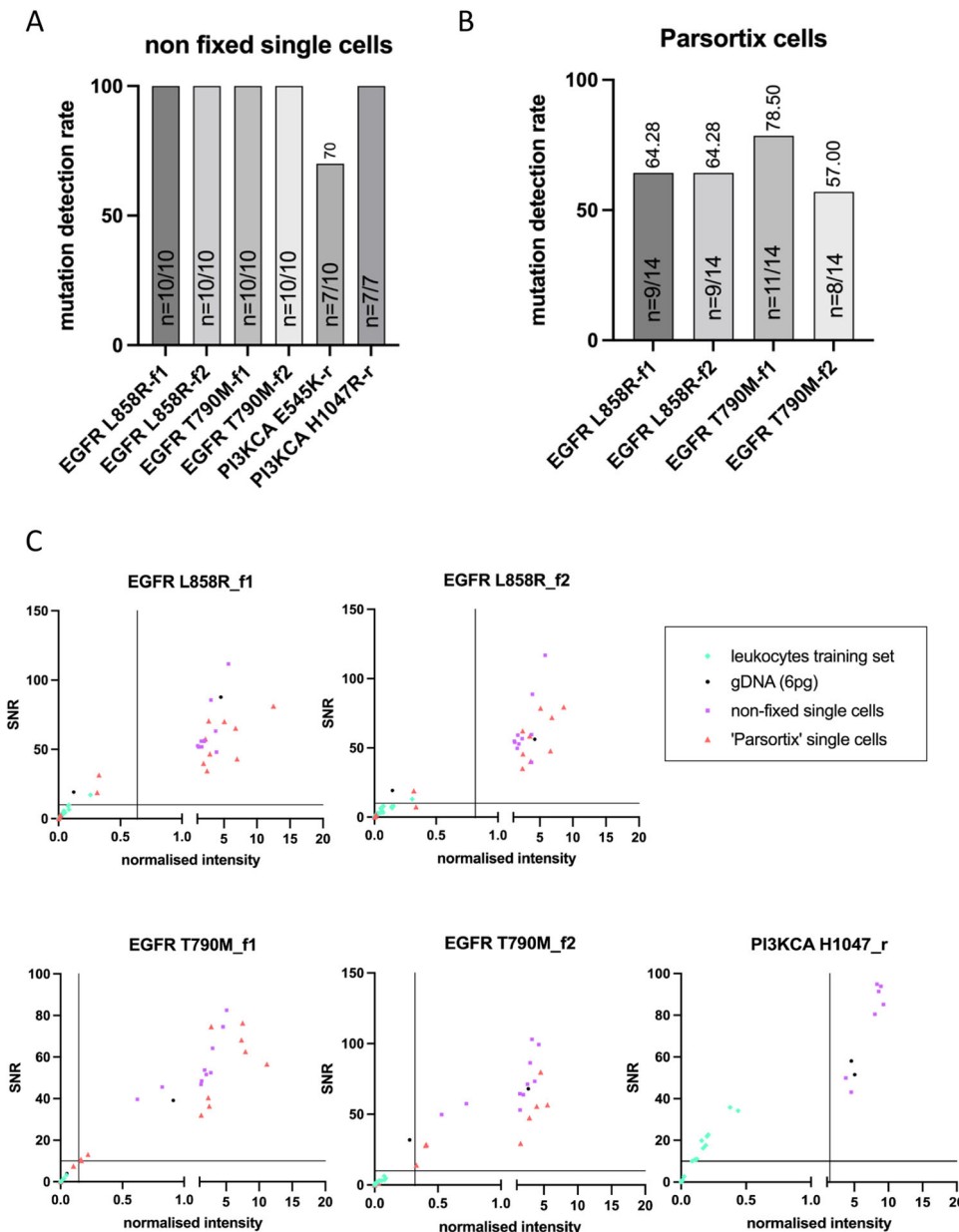

**Figure EV3. Mutation detection at the single cell level with the UltraSEEK® Lung Panel.**

(A) The UltraSEEK® Lung Panel was tested on 2 *EGFR* mutations and 2 *PI3KCA* mutations with our modified protocol on individual cells from lung (H1975) or breast tumor cell lines (MCF7 and T47D) not chemically preserved. The number of cells with successful mutation detection over the number of efficiently lysed cells is mentioned. (B) The UltraSEEK® Lung Panel was tested on the 2 *EGFR* mutations with our modified protocol on individual cells from lung tumor cell lines processed with Parsortix® enrichment method. The number of cells with successful mutation detection over the number of efficiently lysed cells is mentioned. (C) Normalized intensity and signal-to-noise ratio (SNR) values obtained for each PCR assay for gDNA tested at 6.6 pg input in duplicate, not chemically preserved single cells and 'Parsortix' processed single cells. Source data are available online for this figure.

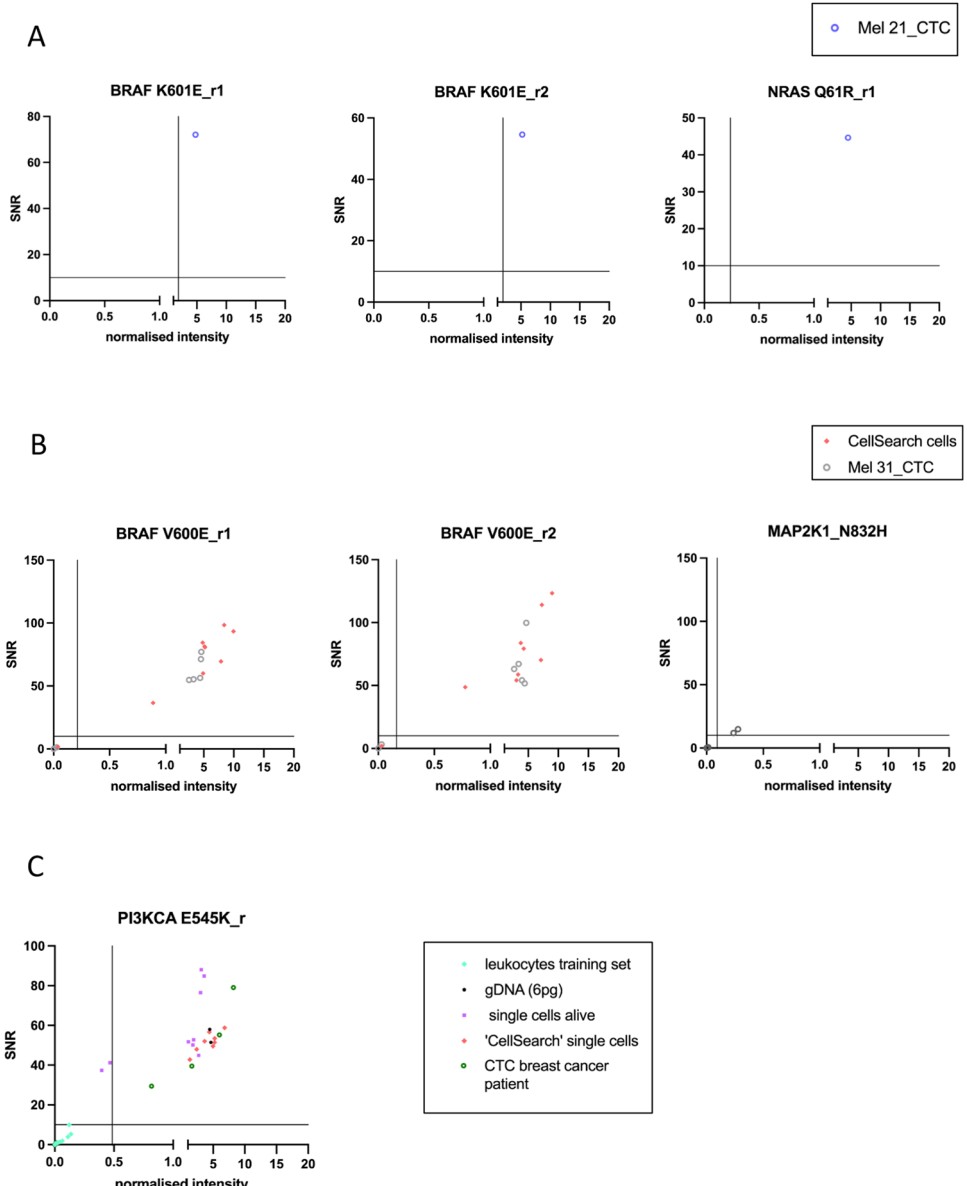

**Figure EV4.   Normalized intensity and SNR values for the mutations detected among CTC processed with the CellSearch® enrichment method.**

For each assay, the horizontal line represents the threshold value of SNR (set to 10 according to manufacturer's recommendation) and of the normalized intensity calculated on leukocytes. (**A**) Normalized intensity and SNR values for the mutations detected among CTC from patient MEL 21. (**B**) Normalized intensity and SNR values for the mutations detected among CTCs from patient MEL 31. (**C**) Normalized intensity and SNR values for the PI3KCA E545K mutation detected among breast tumor cell lines and CTC from breast cancer patient. Source data are available online for this figure.

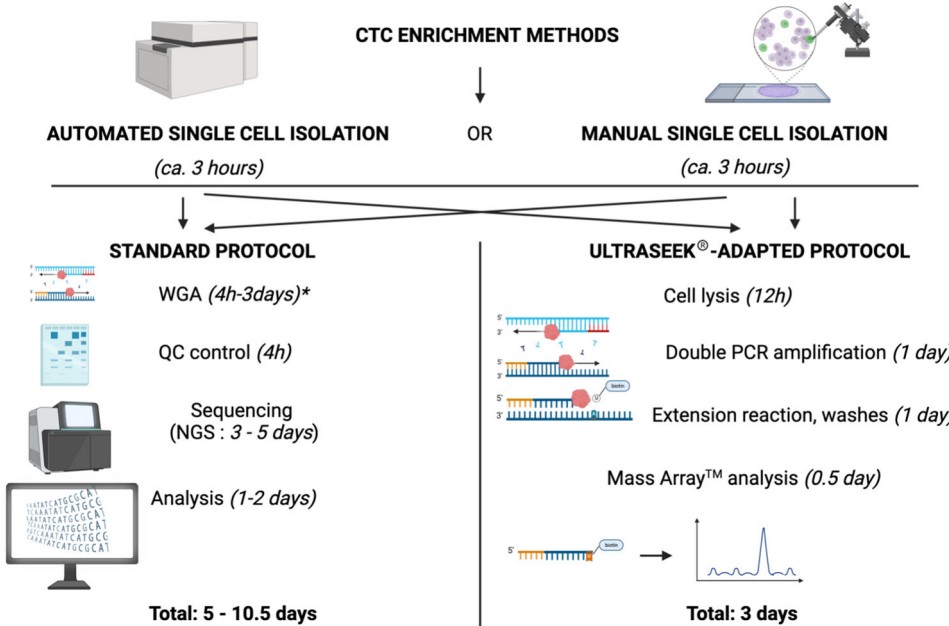

**Figure EV5. Comparison of turnaround times of UltraSEEK®-adapted protocol and "standard" protocol to analyze hotspots mutations from single CTCs.**

*Duration of the protocol for WGA is dependent on WGA technologies. We included cell lysis that is also necessary into WGA protocol global time. We chose to compare the UltraSEEK®-adapted protocol to WGA and next-generation sequencing (NGS) as it appears for us that NGS was the most adequate technology to analyze in parallel multiple hotspot mutations. Created using BioRender.com.

