## [Peer Review File · EMBO Molecular Medicine]

Mutation analysis in individual circulating tumor cells depicts intratumor heterogeneity in melanoma

Mark Sementsov, Leonie Ott, Julian Kött, Alexander Sartori, Amélie Lusque, Sarah Degenhardt, Bertille Segier, Isabel Heidrich, Beate Volkmer, Rudiger Greinert, Peter Mohr, Ronald Simon, Julia Stadler, Darryl Irwin, Claudia Koch, Antje Andreas, Benjamin Deitert, Verena Thewes, Andreas Trumpp, Andreas Schneeweiss, Yassine Belloum, Sven Peine, Harriet Wikman, Sabine Riethdorf, Stefan Schneider, Christoffer Gebhardt, Klaus Pantel, and Laura Keller

Corresponding authors: Laura Keller (keller.laura@iuct-oncopole.fr) , Klaus Pantel (pantel@uke.de)

Review Timeline:

Submission Date:	12th Sep 23
Editorial Decision:	13th Sep 23
Appeal:	13th Sep 23
Editorial Decision:	25th Oct 23
Revision Received:	19th Feb 24
Editorial Decision:	15th Apr 24
Revision Received:	3rd May 24
Accepted:	14th May 24

Editor: Poonam Bheda

Transaction Report:

13th Sep 2023

Decision on your manuscript EMM-2023-18555

Dear Dr. Keller,

Thank you for submitting your manuscript "Mutation analysis in individual circulating tumor cells depicts intratumor heterogeneity in melanoma" to EMBO Molecular Medicine. I have now carefully read your manuscript and discussed it with my colleagues. I regret to say that we all agree that the manuscript is not well suited for publication in EMBO Molecular Medicine and therefore have decided not to proceed with peer review.

In this study, you identify potential driver mutations in single circulating tumor cells (CTCs). We appreciate that analysis of single CTCs can identify a higher number of different mutations than cell-free DNA analysis and can also be used to assess tumor heterogeneity. However, we feel that the advance presented in the work remains limited since the UltraSeek method was previously developed and applied towards intra-patient heterogeneity assessment. Thus, we are unfortunately not persuaded that they provide the sort of conceptual advance we would expect in an EMBO Molecular Medicine article.

I am sorry I could not bring better news regarding the publication of your study in EMBO Molecular Medicine.

Yours sincerely,

Poonam Bheda, PhD
Scientific Editor
EMBO Molecular Medicine

As a service to authors, EMBO provides authors with the possibility to transfer a manuscript that one journal cannot offer to publish to another EMBO publication. The full manuscript and if applicable, reviewers reports are automatically sent to the receiving journal to allow for fast handling and a prompt decision on your manuscript. For more details of this service, and to transfer your manuscript to another EMBO title please click on Link Not Available

Dear Dr. Bheda,

I was just informed by Laura Keller (in cc) about the rejection of our manuscript. I fully agree with your comment that the Ultraseek CtDNA technology has been already published but I think that the key message of our present work might have been overlooked: the conceptual advance is that single cell DNA analysis (of CTCs or any other tumor cells) is now possible without the need of a WGA probe to induce bias. To my best knowledge, this has not been reported before and may open a new avenue. It would be therefore wonderful if we could re-submit a revised version and make this point clearer.

Best wishes,
Klaus Pantel

25th Oct 2023

Dear Dr. Keller,

Thank you for the submission of your manuscript to EMBO Molecular Medicine. We have now received feedback from the three reviewers who agreed to evaluate your manuscript. As you will see from the reports below, the referees acknowledge the interest of the study and are overall supporting publication of your work pending appropriate revisions. Please note that benchmarking to another WGA approach as suggested by Reviewer 3 will not be required for further consideration of your manuscript, however addressing the remaining reviewers' concerns in full will be necessary for further considering the manuscript in our journal, and acceptance of the manuscript will entail a second round of review. EMBO Molecular Medicine encourages a single round of revision only and therefore, acceptance or rejection of the manuscript will depend on the completeness of your responses included in the next, final version of the manuscript. For this reason, and to save you from any frustrations in the end, I would strongly advise against returning an incomplete revision.

We are expecting your revised manuscript within three months, if you anticipate any delay, please contact us.

We require:

4) A .docx formatted letter INCLUDING the reviewers' reports and your detailed point-by-point responses to their comments. As part of the EMBO Press transparent editorial process, the point-by-point response is part of the Review Process File (RPF), which will be published alongside your paper.

5) A complete author checklist, which you can download from our author guidelines (<https://www.embopress.org/page/journal/17574684/authorguide#submissionofrevisions>). Please insert information in the checklist that is also reflected in the manuscript. The completed author checklist will also be part of the RPF.

6) Please note that all corresponding authors are required to supply an ORCID ID for their name upon submission of a revised manuscript.

7) It is mandatory to include a 'Data Availability' section after the Materials and Methods. Before submitting your revision, primary datasets produced in this study need to be deposited in an appropriate public database, and the accession numbers and database listed under 'Data Availability'. Please remember to provide a reviewer password if the datasets are not yet public (see <https://www.embopress.org/page/journal/17574684/authorguide#dataavailability>).

In case you have no data that requires deposition in a public database, please state so in this section. Note that the Data Availability Section is restricted to new primary data that are part of this study. This study includes no data deposited in external repositories.

8) For data quantification: please specify the name of the statistical test used to generate error bars and P values, the number (n) of independent experiments (specify technical or biological replicates) underlying each data point and the test used to calculate p-values in each figure legend. The figure legends should contain a basic description of n, P and the test applied. Graphs must include a description of the bars and the error bars (s.d., s.e.m.). Please provide exact p values.

9) Our journal encourages inclusion of *data citations in the reference list* to directly cite datasets that were re-used and obtained from public databases. Data citations in the article text are distinct from normal bibliographical citations and should directly link to the database records from which the data can be accessed. In the main text, data citations are formatted as

follows: "Data ref: Smith et al, 2001" or "Data ref: NCBI Sequence Read Archive PRJNA342805, 2017". In the Reference list, data citations must be labeled with "[DATASET]". A data reference must provide the database name, accession number/identifiers and a resolvable link to the landing page from which the data can be accessed at the end of the reference. Further instructions are available at .

13) Author contributions: CRediT has replaced the traditional author contributions section because it offers a systematic machine readable author contributions format that allows for more effective research assessment. Please remove the Authors Contributions from the manuscript and use the free text boxes beneath each contributing author's name in our system to add specific details on the author's contribution. More information is available in our guide to authors.

Please also suggest a striking image or visual abstract to illustrate your article as a PNG file 550 px wide x 300-600 px high. Share synopsis text and image, as well as eTOC:

Please note that these would be the final versions and changes during proofing are usually not allowed

16) As part of the EMBO Publications transparent editorial process initiative (see our Editorial at <http://embomolmed.embopress.org/content/2/9/329>), EMBO Molecular Medicine will publish online a Review Process File (RPF) to accompany accepted manuscripts.

In the event of acceptance, this file will be published in conjunction with your paper and will include the anonymous referee reports, your point-by-point response and all pertinent correspondence relating to the manuscript. Let us know whether you agree with the publication of the RPF and as here, if you want to remove or not any figures from it prior to publication.

I look forward to receiving your revised manuscript.

Yours sincerely,

Poonam Bheda

Poonam Bheda, PhD
Scientific Editor
EMBO Molecular Medicine

**** Reviewer's comments ****

Referee #1 (Remarks for Author):

This manuscript addresses an important critical bottleneck in the characterization of CTCs. The implementation of which may help improve the reliability and reproducibility of mutation analyses from CTCs. The ability to identify more clonal/subclonal events and mutational heterogeneity in CTCs compared with paired tumor and ctDNA will help researchers understand the metastatic process/niche better. Overall an excellent technically innovative manuscript which is succinctly written.

Nevertheless limitations of manual picking of CTCs and turnaround time of such techniques may limit to research use only utility, and the reader would like to hear the authors view on this.

Referee #2 (Comments on Novelty/Model System for Author):

Please see the remarks for the authors.

Referee #2 (Remarks for Author):

This is an emerging technology which may significantly facilitate the mutation-based clinical diagnosis of cancers. Therefore, the QC process needs to be examined.

1. Please compare the workflow in this study with that in conventional NGS workflow in a table/figure and note the turnover time. Noticeably, both methods have the same bottleneck in CTC capturing.
2. A double-blind test of negative control should be performed.
3. What is the detection limit of both CTC capturing methods (e.g. number of CTC/ml of blood)?
4. Many melanomas express low levels of melanocytic genes used as melanoma markers (e.g. MLANA, PMEL, S100). In that case, an alternative approach is to enrich melanoma cells by negative selection; e.g. depleting CD45+, CD31+, and CD34+ cells. This may solve the issue of inconsistent sampling. If possible, please test it.

Referee #3 (Remarks for Author):

Reference: EMM-2023-18555-V2-Q

Title: Mutation analysis in individual circulating tumour cells depicts intratumor heterogeneity in melanoma

Reviewers response:

Keller and colleagues describe the genomic analysis of single circulating tumour cells (CTCs) to address questions of intra-patient heterogeneity in metastatic melanoma.

The study describes modification of a mass spectrometry based approach of mutation detection from individual CTCs not requiring WGA and complex bioinformatics pipelines. The first part of the manuscript describes establishment of the protocol on single cells of cell lines to show sensitivity and reproducibility of the technique. The approach is then tested on a 3 clinical samples and then a cohort of 33 metastatic melanoma patients with CTC mutations identified compared to those obtained from

tumour tissue and ctDNA.

The study is interesting but does have a number of limitations:

1. Small and mixed set of samples and experiments with variable results: The study utilises a mixture of cell lines and clinical samples, which are processed and analysed a number of different ways. The authors then combine these approaches to confirm the reproducibility of the analysis. However, this makes the manuscript rather confusing as it goes from melanoma to breast cancer, CellSearch/DEPArray analysis to Parsortix/single cell picking with no clear comparison across.

- for example the NRASQ61R concomitant mutation seen in MEL21 and MAP2K1 mutation in MEL31 (Figure 2) are not seen in the Parsortix data presented later (Figure 3) but not discussed. Variability in CTC mutations is an issue throughout the study which is attributed to heterogeneity, but in my opinion technical reliability is a concern that the manuscript does not address.

2. The technical reliability of the approach: The authors state that an advantage of the UltraSEEK approach is that it does not rely on WGA and so it introduces fewer errors so should be more reliable. However there is considerable variability in the CTC data presented across the clinical samples. This could be attributed to a number of possibilities:

- true tumour heterogeneity
- isolated CTCs not actually true tumour derived cells
- technical noise within approach

I think all three of these play a role in the results, could the authors address these issues? Ideally the UltraSEEK approach should be bench marked to another WGA approach, though this maybe beyond the scope of the current study? Can the authenticity of the CTCs be proven - low pass CNA analysis of cells for example? Data from patient leukocytes showing WT calls across all mutations? Where ctDNA/tissue mutations and CTC data does not match what are the VAF% of the missed mutations, in the data presented in Figure 3 is it difficult to see a pattern of clonal and sub-clonal mutations being pick up in the patients?

3. Applicability of approach: The authors state that "...we obtained efficient rates of single cell lysis and DNA amplification (superior to 70% in CTC from patients) and recovery rates (superior to 70%) for several mutations on CTCs isolated by the two FDA-cleared methods" which suggests the approach is robust and reliable. However, for the metastatic melanoma cohort the authors successfully analysed 132/250 CTC, of which 60 had mutations. This is a detection level of <25%? I feel a clearer appraisal of the technical difficulties and limitations of the approach is needed within the discussion.

Dear Dr. Poonam Bheda,

We are delighted to learn that our manuscript entitled „Mutation analysis in individual circulating tumour cells depicts intratumor heterogeneity in melanoma“ (ID EMM-2023-18555-V2) has been evaluated positively and we acknowledge the possibility to resubmit a revised version of our manuscript.

We used the comments of the reviewers to critically discuss our study. We have now extensively revised the manuscript, according to the reviewer's suggestions to make it clearer and more consistent for the reader. We added a number of additional analyses, which have improved the technical reliability assessment of our protocol.

All in all, we feel that the manuscript quality has considerably improved by the revision.

A detailed point-to-point response to the comments of the reviewers is attached below, including a version where all changes and additions are highlighted in blue.

We are looking forward to your response.

Sincerely yours,

Pr. Christoffer Gebhardt

Pr. Klaus Pantel

Dr. Laura Keller

Point-to-point response:

Referee #1

Remarks for Authors:

This manuscript addresses an important critical bottleneck in the characterization of CTCs. The implementation of which may help improve the reliability and reproducibility of mutation analyses from CTCs. The ability to identify more clonal/subclonal events and mutational heterogeneity in CTCs compared with paired tumor and ctDNA will help researchers understand the metastatic process/niche better. Overall an excellent technically innovative manuscript which is succinctly written.

Nevertheless limitations of manual picking of CTCs and turnaround time of such techniques may limit to research use only utility, and the reader would like to hear the authors view on this.

Response:

We thank the Reviewer for the overall very positive evaluation and appreciate his constructive specific feedback. We now critically discuss the limitations on manual picking and the need for more automated and efficient methods to isolate single cells and for their molecular characterization (line 351 p 14): *“Single-cell isolation is the first bottleneck that needs to be further improved by more efficient automated methods like the DepArray[®] used in our study for single cell isolation of tumor cell-line cells and CTCs isolated by the CellSearch[®] system. Even if we managed to pick 73.2% of the potential CTC we detected, the workflow used for the melanoma cohort relies on manual picking whose success rate can be dependent on user’s expertise.”*

We also have included estimation of the turnaround time of the proposed workflow, which was only about 3 days in comparison to WGA followed by NGS analysis that can usually be accomplished in 5 to 10 days, line 283 p 12 and Figure EV 5: *“After CTC isolation, standard workflow typically requires WGA followed by NGS analysis and usually takes about 5 to 10 days depending on the protocols used; here, we report on the successful development of a simplified workflow that amplifies the regions of interest, allowing the interrogation of multiple hotspot mutations at the single cell level without requiring prior WGA, does not depend on NGS and that can be performed in 3 days, therefore presenting a substantial gain in the turn-around time of the analysis”* Turnaround time is an important parameter for implementing a technology into clinical decision making, in particular in patients with advanced metastatic disease who need a rapid decision on their therapies of choice.

Figure EV 5: Comparison of turnaround times of UltraSEEK[®]-adapted protocol and “standard” protocol to analyse hotspot mutations from single CTC.

* duration of the protocol for WGA is dependent on WGA technologies. We included cell lysis that is also necessary into WGA protocol global time. We chose to compare the UltraSEEK[®]-adapted protocol to WGA and Next Generation Sequencing (NGS) as it appears for us that NGS was the most adequate technology to analyse in parallel multiple hotspot mutations.

Referee #2

Remarks for Authors:

This is an emerging technology which may significantly facilitate the mutation-based clinical diagnosis of cancers. Therefore, the QC process needs to be examined.

1. Please compare the workflow in this study with that in conventional NGS workflow in a table/figure and note the turnover time. Noticeably, both methods have the same bottleneck in CTC capturing.

Response: We revised our discussion to include a more specific comparison with conventional NGS workflow, as suggested by the reviewer in Figure EV 5. We also have added an estimation of turnover time (line 282 p12): *“After CTC isolation, standard workflow typically requires WGA followed by NGS analysis and usually takes about 5 to 10 days depending on the protocols used; here, we report on the successful development of a simplified workflow that amplifies the regions of interest, allowing the interrogation of multiple hotspot mutations at the single cell level without requiring prior WGA, does not depend on NGS and that can be performed in 3 days, therefore presenting a substantial gain in the turn-around time of the analysis”*

We agree with the reviewer that both molecular characterization workflows (conventional NGS or ours) are limited by the CTC capture rate, and added a comment to the Discussion line 392 p16:

“However, molecular characterization of CTCs is limited by the CTC capture rate, independent from the downstream molecular assay used for genomic profiling.”

We also critically discuss the limitations on manual picking and the need for more automated and efficient methods to isolate single cells, see response to Reviewer 1, (line 351 p 14): *“Single-cell isolation is the first bottleneck that needs to be further improved by more efficient automated methods like the DepArray[®] used in our study for single cell isolation of tumor cell-line cells and CTCs isolated by the CellSearch[®] system. Even if we managed to pick 73.2% of the potential CTC we detected, the workflow used for the melanoma cohort relies on manual picking whose success rate can be dependent on user’s expertise.”*

2. A double-blind test of negative control should be performed.

Response:

We thank the reviewer for suggesting this experiment that has allowed us to improve the specificity of our results. We tested 12 and 5 leukocytes for the UltraSEEK Melanoma and Lung Panels respectively (Tables EV 2 and 7). For the lung panel, we did not detect any positive signals in leukocytes. For the melanoma panel, 7/76 PCR assays (corresponding to 6 mutations) presented positive signals in leukocytes, while we did not detect any positive signals in the others 69 assays (lines 132 p5): *“In order to validate the mutation calling process (based on the normalized intensity and the SNR value), we defined a validation set composed of 15 single cells from 2 different cell lines (A2058, SKMEL30) and 12 leukocytes from one healthy donor. We observed 11 positive calls in 7 different mutations assays (CTNNB1_S45P-f1 ; KIT_V559A-f1 ; KIT_V559A-f2 ; KIT_L576P-f2 ; MAP2K1_P124S-f2; SDHD ‘mut1’-f1 ; SDHD ‘mut2’-f1, Table EV 2) among the leukocytes. In melanoma tumor cell lines, we did not detect any other mutation apart from the expected BRAF V600E, MAP2K1 P124S (f1) and NRAS Q61K mutations (Table EV 2). While we have not observed any unexpected mutation in 90.8% (69/76) of the assays, 6.6% (5/76) of the assays (CTNNB1_S45P-f1; KIT_L576P-f2; MAP2K1_P124S-f2; SDHD_mut1-f1; SDHD_mut2-f1), present a rate of unexpected mutation of 3.7% (1/27) and 2.63% (2/76) of the assays (KIT_V559A-f1 ; KIT_V559A-f2) present a rate of unexpected mutation of 11.1% (3/27). Therefore, the mutation calling strategy is validated for the vast majority of the UltraSEEK[®] Melanoma Panel assays and the assays with positive values on leukocytes were excluded from further analysis.”* We decided to exclude these assays from patient analysis and updated the descriptive results accordingly (line 183 p7): *“Seven assays that presented positive signals in leukocytes were excluded from the analysis of mutations in patients and affected 5 CTCs. Overall, we found mutations in 55.6% (15/27) of the samples corresponding to 57/132 (43.2%) of CTCs efficiently lysed. In the vast majority (52/57, 91.2%) of the CTCs, only one mutation was found, while 5/57 (8.8%) CTCs had 2 mutations or more”.* We also now critically discuss the analytical performance of our protocol (line 350 p14): *“Second, mutation detection from single CTCs remains technically challenging at different steps. Single-cell isolation is the first bottleneck that needs to be further improved by more efficient automated methods automated methods like the DepArray used in our study for single cell isolation of tumor cell-line cells and CTCs isolated by the CellSearch system. Even if we managed to pick 73.2% of the potential CTC we detected, the workflow used for the melanoma cohort relies on manual picking whose success rate can be dependent on user’s expertise. Lysis efficiency is the next step that can lead to some loss even if we managed to lyse more than 70% of patient CTCs and finally, the mutation detection efficiency of our protocol is around 81.5% (among lysed cells) based on the results of our tumor cell lines experiments. Therefore, the combination of these different technical steps impact our mutation detection efficiency from CTC. Nevertheless, the observed detection rate of mutations in CTCs of this cohort (43%, 57/132) could be as well explained by the limited number of mutations interrogated by the panel (albeit designed to detect the most frequent ones) and biological factors such as the unknown percentage of cells within each tumor bearing the mutation and the fact that CTCs are tumor cells selected through the tumor dissemination*

process. Consequently, all these parameters play a role in the appraisal of intra-tumor heterogeneity and in the comparison of the molecular content of CTC with tumor tissue and/or ctDNA. Specificity of our mutation calling approach also plays a role in that matter. Our mutation calling strategy is based on two parameters (normalized intensity and SNR) that reflect two complementary characteristics of the peak. While the vast majority (90.8%) of the assays did not present any signal in leukocytes used as negative control, we observed positive signals in some leukocytes for 7 assays that we excluded from CTC analysis.”

3. What is the detection limit of both CTC capturing methods (e.g. number of CTC/ml of blood)?

Response: According to Templeman *et al.* (PMID: 37601320), the limit of detection of the Parsortix[®] method varies between 3-5 cells (depending on the cell line used). The limit of detection of the CellSearch[®] platform is 1 epithelial cell according to the user guide. Allard *et al.* (PMID: 15501967) have performed an extensive analytical validation of the CellSearch[®] platform. Considering a recovery rate of 85% (verified experimentally across different number of cells spiked), it is necessary that a sample contains 1.2+/-0.4 CTC for one CTC to be detected in a sample of 7.5ml of blood.

4. Many melanomas express low levels of melanocytic genes used as melanoma markers (e.g. MLANA, PMEL, S100). In that case, an alternative approach is to enrich melanoma cells by negative selection; e.g. depleting CD45+, CD31+, and CD34+ cells. This may solve the issue of inconsistent sampling. If possible, please test it.

Response: In this study, after Parsortix[®] enrichment (which is a label-independent CTC enrichment method), melanoma CTCs were detected as nucleated elements negative for CD45 and CD16 but either positive for the membrane antigens MCAM or NG2 or S100B and for melanocytic genes like MART-1, GP100, Tyrosinase. That is why MCAM and NG2 were assessed in a different channel from melanocytic genes in our immunofluorescence assay. We believe that this strategy could help to detect melanoma CTCs in case of low expression of melanocytic antigens. Indeed, we were able to detect melanoma CTCs either with high expression of MCAM/NG2 antigens and melanocytic antigens (see Figure EV 2B) or also with high MCAM/NG2 expression but low expression of melanocytic antigens, as shown below in two additional examples, now also incorporated into Figure EV 2B. Thus, we feel that our strategy captures the known heterogeneity of melanocytic gene expression on CTCs quite well.

We agree with the reviewer that negative selection by depletion of CD45 positive cells could be an alternative strategy in case of low levels of antigens in CTCs. However, negative depletion strategies usually result in lower purity, which makes it more difficult to find the single tumor cells in a higher background of residual leukocytes. Moreover, to identify the melanoma cells at the single cell level,

we still need to employ antibodies against tumor-associated antigens and low antigen expression may be then also a source of missing CTCs.

Referee #3

Keller and colleagues describe the genomic analysis of single circulating tumour cells (CTCs) to address questions of intra-patient heterogeneity in metastatic melanoma.

The study describes modification of a mass spectrometry based approach of mutation detection from individual CTCs not requiring WGA and complex bioinformatics pipelines. The first part of the manuscript describes establishment of the protocol on single cells of cell lines to show sensitivity and reproducibility of the technique. The approach is then tested on a 3 clinical samples and then a cohort of 33 metastatic melanoma patients with CTC mutations identified compared to those obtained from tumour tissue and ctDNA.

The study is interesting but does have a number of limitations:

1. Small and mixed set of samples and experiments with variable results: The study utilises a mixture of cell lines and clinical samples, which are processed and analysed a number of different ways. The authors then combine these approaches to confirm the reproducibility of the analysis. However, this makes the manuscript rather confusing as it goes from melanoma to breast cancer, CellSearch/DEPArray analysis to Parsortix/single cell picking with no clear comparison across. - for example the NRASQ61R concomitant mutation seen in MEL21 and MAP2K1 mutation in MEL31 (Figure 2) are not seen in the Parsortix data presented later (Figure 3) but not discussed. Variability in CTC mutations is an issue throughout the study which is attributed to heterogeneity, but in my opinion technical reliability is a concern that the manuscript does not address.

Response: We acknowledge that the presentation of our validation method might be confusing. Our aim was not to compare two workflows (Parsortix[®] + manual picking vs. CellSearch[®] + DepArray[®]) but rather to simply illustrate the applicability of our protocol with the two commonly used workflows for CTC enrichment and isolation that have been approved by FDA for some solid tumors. In order to improve the consistency of our manuscript, we have grouped together in the first figure the experiments related to analysis of mutations from 'Parsortix[®] tumor cells' or 'Parsortix[®] CTC' with the UltraSEEK[®] Melanoma Panel. In the main text we now focus on the work on the melanoma cohort, and present separately proof-of-principle data on tumor cell lines and patient CTCs related to a potential applicability of this protocol to (a) other CTC enrichment methods like the EpCAM-dependent CellSearch[®] and (b) other clinical entities like lung or breast cancer (see new Figure 2 and new chapter line 250 p10 "*Application of the modified UltraSEEK[®] protocol to other cancer entities and other CTC enrichment methods*").

We also now critically discuss the detection of mutations within CellSearch[®] CTC and not within Parsortix[®] CTC from the two melanoma cases, line 299 p12: "*In the two melanoma cases where CTC were both isolated with CellSearch or Parsortix enrichment methods, the driver mutation present in the tumor tissue was detected on CTC enriched with both methods. Nevertheless, we also report mutations exclusively found in the 'CellSearch-CTC' that were not found in the 'Parsortix-CTC'. This can be explained by the fact that both methods capture (different) CTCs based on different enrichment principles (membrane antigen for CellSearch and size/deformability for Parsortix), which might explain the heterogeneity of results.*"

Technical reliability may refer to sensitivity (i.e., detection rate of mutation known to be present in a cell line) and specificity (i.e., no detection of mutation in negative samples). We provided data regarding sensitivity by assessing some of the most frequent mutations found in melanoma or lung cancer in tumor cell lines processed through Parsortix or CellSearch platforms, as the most representative model of our application on patient CTCs. We have completed our assessment of sensitivity with the Parsortix platform by testing 8 SKMEL2 (carrying the NRAS Q61R mutation, detection rate: 87.5%, Figure 1A) cells with the melanoma panel and 10 additional H1975 cells (carrying the EGFR T790M and L858R mutations, Figure EV3B) with the lung panel. To assess specificity, we initially tested our protocol on 15 single tumor cells on the entire melanoma panel and on 4 single tumor cells for the entire lung panel and did not observe any additional mutation except the ones expected to be present (Tables EV 2 and 7). We now further assessed the specificity by testing 12 and 5 additional leukocytes on the melanoma and lung panel, respectively. For the lung panel, we did not detect any positive signals in leukocytes. For the melanoma panel, 7/76 PCR assays (corresponding to 6 mutations) presented positive signals while we did not detect any positive signals in the others 70 assays (lines 132 p5): *“In order to validate the mutation calling process (based on the normalized intensity and the SNR value), we defined a validation set composed of 15 single cells from 2 different cell lines (A2058, SKMEL30) and 12 leukocytes from one healthy donor. We observed 11 positive calls in 7 different mutations assays (CTNNB1_S45P-f1 ; KIT_V559A-f1 ; KIT_V559A-f2 ; KIT_L576P-f2 ; MAP2K1_P124S-f2; SDHD ‘mut1’-f1 ; SDHD ‘mut2’-f1, Table EV 2) among the leukocytes. In melanoma tumor cell lines, we did not detect any other mutation apart from the expected BRAF V600E, MAP2K1 P124S (_f1) and NRAS Q61K mutations (Table EV 2). While we have not observed any unexpected mutation in 90.8% (69/76) of the assays, 6.6% (5/76) of the assays (CTNNB1_S45P-f1; KIT_L576P-f2; MAP2K1_P124S-f2; SDHD_mut1-f1; SDHD_mut2-f1), present a rate of unexpected mutation of 3.7% (1/27) and 2.63% (2/76) of the assays (KIT_V559A-f1 ; KIT_V559A-f2) present a rate of unexpected mutation of 11.1% (3/27). Therefore, the mutation calling strategy is validated for the vast majority of the UltraSEEK® Melanoma Panel assays and the assays with positive values on leukocytes were excluded from further analysis.”* We decided to exclude these PCR assays from patient analysis and updated the descriptive results accordingly (line 183 p7): *“Seven assays that presented positive signals in leukocytes were excluded from the analysis of mutations in patients and affected 5 CTCs. Overall, we found mutations in 55.6% (15/27) of the samples corresponding to 57/132 (43.2%) of CTCs efficiently lysed. In the vast majority (52/57, 91.2%) of the CTCs, only one mutation was found, while 5/57 (8.8%) CTCs had 2 mutations or more”*. Even without those PCR assays, the number of mutations within CTC remains higher than in ctDNA (Figure 1B). We agree with the reviewer that these analytical performances should be discussed. We therefore added the following statements (line 350 p14): *“Second, mutation detection from single CTC remains technically challenging at different steps. Single-cell isolation is the first bottleneck that needs to be further improved by more efficient automated methods like the DepArray used in our study for single cell isolation of tumor cell-line cells and CTCs isolated by the CellSearch system. Even if we managed to pick 73.2% of the potential CTC we detected, the workflow used for the melanoma cohort relies on manual picking whose success rate can be dependent on user’s expertise. Lysis efficiency is the next step that can lead to some loss even if we managed to lyse more than 70% of patient CTCs and finally, the mutation detection efficiency of our protocol is around 81.5% (among lysed cells) based on the results of our tumor cell lines experiments. Therefore, the combination of these different technical steps impact our mutation detection efficiency from CTC. Nevertheless, the observed detection rate of mutations in CTCs of this cohort (43%, 57/132) could be as well explained by the limited number of mutations interrogated by the panel (albeit designed to detect the most frequent ones) and biological factors such as the unknown percentage of cells within each tumor bearing the mutation and the fact that CTCs are tumor cells selected through the tumor dissemination process. Consequently, all these parameters play a role in the appraisal of intra-tumor heterogeneity and in the comparison of the molecular content of CTC with tumor tissue and/or ctDNA. Specificity of our mutation calling approach also plays a role in that matter. Our mutation calling strategy is based on two parameters (normalized intensity and SNR) that reflect two complementary characteristics of the peak. While the*

vast majority (90.8%) of the assays did not present any signal in leukocytes used as negative control, we observed positive signals in some leukocytes for 7 assays that we excluded from CTC analysis."

2. The technical reliability of the approach: The authors state that an advantage of the UltraSEEK approach is that it does not rely on WGA and so it introduces fewer errors so should be more reliable. However there is considerable variability in the CTC data presented across the clinical samples. This could be attributed to a number of possibilities:

- true tumour heterogeneity
- isolated CTCs not actually true tumour derived cells
- technical noise within approach

I think all three of these play a role in the results, could the authors address these issues?

Response: (i) True tumor heterogeneity has been already discussed from lines 311 p13 to 346 p14.

(ii) To reduce the chance that isolated CTCs are not actually true tumor derived cells, we established an immunofluorescence staining protocol for detection of melanoma CTCs that includes "negative exclusion" markers expressed on leukocytes, which is currently state-of-the-art in CTC detection (Figure EV 2B).

(iii) Technical noise is now also discussed line 350 p14 (see above reply to point 1).

Ideally the UltraSEEK approach should be benchmarked to another WGA approach, though this may be beyond the scope of the current study?

Response: We agree with the reviewer that a comprehensive side-by-side comparison is beyond the scope of this study and would also require substantial higher blood draw volumes since we are dealing with the detection of rare events (ie., CTCs). However, as proof-of-principle we used WGA and Sanger sequencing to confirm the PI3KCA mutation detected in the breast cancer patient with the modified UltraSEEK[®] protocol.

Can the authenticity of the CTCs be proven - low pass CNA analysis of cells for example?

Response: We used all cell lysate into the modified UltraSEEK[®] approach and can therefore not assess tumor origin of CTCs by low pass CNA analysis. Nevertheless, we have used a combination of positive markers and exclusion markers to reduce the chance that the CTCs analysed in this work were non-tumor cells circulating in the blood (see reply to point 4 of reviewer N2 and Figure EV2B).

Data from patient leukocytes showing WT calls across all mutations?

Response: The threshold established from leukocytes in the training set for all mutations in the panel is provided in Tables EV 2 and 7, and we have added now another table with normalised intensity value of each leukocyte included in the training set (Table EV1 and 6). Normalised intensity values and SNR have also been added to Tables EV 2 and 7 for the leukocytes "validation set".

Where ctDNA/tissue mutations and CTC data does not match what are the VAF% of the missed mutations, in the data presented in Figure 3 is it difficult to see a pattern of clonal and sub-clonal mutations being picked up in the patients?

Response: The VAF for ctDNA mutations are presented in Table EV 4 and for tissue in Table EV 5. Please note that the ctDNA UltraSEEK[®] protocol only provides semi-quantitative estimation of VAF

and the Mass Array SVR output does not report estimation above than 2%. For calling mutations in CTC, we do not consider the VAF calculated by the software but rather the normalised intensity (represented by the 'allele frequency' column in the SVR output) of the mutation peak in the mass spectrum, and the signal-to-noise ratio of the peak. In the data presented in Figure 1C, we agree with the reviewer that a pattern of clonal and sub-clonal mutations is mostly visible in patients with a high number of CTC like for patients MEL30, 31 and 32. In these cases, the mutation detected in the tissue and/or ctDNA is also the one most frequent one found in CTCs.

3. Applicability of approach: The authors state that "...we obtained efficient rates of single cell lysis and DNA amplification (superior to 70% in CTC from patients) and recovery rates (superior to 70%) for several mutations on CTCs isolated by the two FDA-cleared methods" which suggests the approach is robust and reliable. However, for the metastatic melanoma cohort the authors successfully analysed 132/250 CTC, of which 60 had mutations. This is a detection level of <25%? I feel a clearer appraisal of the technical difficulties and limitations of the approach is needed within the discussion.

Response: We thank the author for this comment and have significantly revised the discussion section by explaining the technical difficulties of each step of mutation detection from CTC from line 350 p 14 (see reply to point 1).

"Second, mutation detection from single CTC remains technically challenging at different steps. Single-cell isolation is the first bottleneck that needs to be further improved by more efficient automated methods like the DepArray used in our study for single cell isolation of tumor cell-line cells and CTCs isolated by the CellSearch system. Even if we managed to pick 73.2% of the potential CTC we detected, the workflow used for the melanoma cohort relies on manual picking whose success rate can be dependent on user's expertise. Lysis efficiency is the next step that can lead to some loss even if we managed to lyse more than 70% of patient CTCs and finally, the mutation detection efficiency of our protocol is around 81.5% (among lysed cells) based on the results of our tumor cell lines experiments. Therefore, the combination of these different technical steps impact our mutation detection efficiency from CTC. Nevertheless, the observed detection rate of mutations in CTCs of this cohort (43%, 57/132) could be as well explained by the limited number of mutations interrogated by the panel (albeit designed to detect the most frequent ones) and biological factors such as the unknown percentage of cells within each tumor bearing the mutation and the fact that CTCs are tumor cells selected through the tumor dissemination process. Consequently, all these parameters play a role in the appraisal of intra-tumor heterogeneity and in the comparison of the molecular content of CTC with tumor tissue and/or ctDNA."

Our present work was focused on the establishment and validation of a novel method for downstream molecular analysis of CTCs. In the future, more efficient CTC capture and single cell isolation methods will become available and the panel can be extended which will increase the mutation detection rate. Our present manuscript has opened a new avenue for such studies. We accordingly added the following statement line 391 p16: *"However, molecular characterization of CTCs is limited by the CTC capture rate, independent from the downstream molecular assay used for genomic profiling. Technologies to capture higher CTC numbers and to improve the sorting of single CTCs for downstream analyses are currently being established (Keller & Pantel, 2019) which, in combination with the presented platform for genomic single cell analysis may open new avenues for future investigation and potentially envision clinical applications."*

To all referees:

We noticed that our presentation of parameters used to call mutations ('normalised intensity' and 'SNR') should not be presented in two separated graphs but rather plotted together as both contain information relevant to the same individual cell. We have accordingly modified our plots.

After careful inspection of our data on leukocytes analyzed with the UltraSEEK[®] Lung Panel, we have noticed errors in the normalized intensity values from 3 leukocytes included into the training set used to calculate the normalized intensity threshold. New values for the threshold are presented now in Table EV 7. This correction impacts neither the specificity performance of the method, nor the mutations detected from the breast cancer patient. However, this has led to successful detection of the EGFR T790M mutation with the EGFR T790M-f1 assay in 2 H1975 cells (Figure EV 3B,C).

15th Apr 2024

Dear Dr. Keller,

Thank you for the submission of your revised manuscript to EMBO Molecular Medicine. We have now received the enclosed reports from the referees that were asked to re-assess it. As you will see the reviewers are now globally supportive and I am pleased to inform you that we will be able to accept your manuscript pending the following final amendments:

- 1) We note that there are 3 shared senior authorships. Please rename this to "corresponding authors" and provide an institutional email address for contact in the main manuscript file.
- 2) Please the Data availability statement to: "This study includes no data deposited in external repositories."
- 3) Please rename "Competing Interests" to "Disclosure and competing interests statement". We updated our journal's competing interests policy in January 2022 and request authors to consider both actual and perceived competing interests. Please review the policy <https://www.embopress.org/competing-interests> and update your competing interests if necessary.
- 4) In the Materials and Methods, please take care of the following:
 - Although the UltraSEEK Melanoma assay for ctDNA has already been previously described, please include some details of the method and mass spectrometry in the current manuscript.
 - Human research participants: Please include a statement that the experiments conformed to the principles set out in the WMA Declaration of Helsinki and the Department of Health and Human Services Belmont Report.
 - Cell lines: Please be sure to include a sentence in the Materials and Methods as to whether or not the cell lines were recently authenticated and tested for mycoplasma contamination.
 - Please ensure that a statement on whether or not blinding was done is included in the Materials and Methods even if no blinding was done. Please also update the Author Checklist to indicate where this statement was added.
 - Antibodies: please ensure that the dilutions/amounts of each antibody are reported, as these are currently missing.
- 5) Please place individual sections of the manuscript in the following order: Title page - Abstract & Keywords - Introduction - Results - Discussion - Materials & Methods - Data Availability - Acknowledgements - Disclosure and Competing Interests Statement - The Paper Explained - For More Information - References - Figure Legends - Expanded View Figure Legends.
- 6) For the figures and figure legends, please take care of the following:
 - Please remove all figures from main manuscript file and leave only main figure legends placed after the references. Main figures should be uploaded as individual, high res figure files in EPS, TIF or PDF format. Expanded View figures should be renamed "Figure EV1" etc. and should also be uploaded as individual, high resolution figure files. Their legends should remain in the manuscript, after the main figure legends, under the heading "Expanded View Figure Legends" . Please check "Author Guidelines" for more information: <https://www.embopress.org/page/journal/17574684/authorguide#figureformat>
 - Please make sure to update the callouts of all Expanded View figures in the main manuscript text - currently these are missing for Figure EV1, EV4 and EV5. Callouts will also need to be added for the Appendix Tables and Figures.
 - Please note that scale bar and its definition are missing for figures 2b-d.
- 7) Tables: Please upload all tables as one .xsl file per table. Please rename Tables EV0-4 and 6-7 as "Dataset EV1" etc. should be uploaded as individual files. Table EV5 should be renamed Table EV1 and also uploaded as an individual file. Please update their callouts in the main manuscript text.
- 8) Appendix file: Please upload the Appendix file in PDF format. The tables should be named "Appendix Table S1" etc. Suppl. Methods should be moved to the main manuscript and merged with the Materials & Methods. The figure should be named "Appendix Figure S1". The appendix file needs a Table of Contents, including page numbers. Please ensure the word "Appendix" is included in all labels for Appendix Figures and Appendix Tables including in the Table of Contents. Please ensure to update the callouts for each of the figures and tables in the main text.
- 9) Synopsis:
 - Synopsis image: Please resize the synopsis image to 550 pixels wide x (250-400) pixels high and ensure that the figure is uploaded as a high-resolution jpeg file.
 - Synopsis text: Please edit the standfirst to the passive voice (ensure it remains a maximum of 300 characters, including spaces). Please also remove all instructions from the synopsis text file.
 - Please check your synopsis text and image before submission with your revised manuscript. Please be aware that in the proof stage minor corrections only are allowed (e.g., typos).
- 10) Source Data: It is unclear why the microscopy image files for Figures 2B, 2C, and 2D cannot be provided as requested for Source Data. Please provide these files with an updated completed Source Data checklist or explain further as to why these images cannot be provided. Please also ensure that the Source Data structure is uploaded as a single source data file (zipped) per figure, with the panels clearly visible in the folder structure (and not all in a single file).
- 11) The Paper Explained: Please remove "The Paper Explained" from the synopsis text file and add it to the main manuscript text. In addition, please reformat to include the heading "PROBLEM" when describing the medical issue you are addressing, the heading "RESULTS" when discussing the results obtained, and the heading "IMPACT" when discussing their clinical impact. Please see "Author Guidelines" for further details:
<https://www.embopress.org/page/journal/17574684/authorguide#researcharticleguide>
- 12) For more information: This space should be used to list relevant web links for further consultation by our readers. Could you identify some relevant ones and provide such information as well? Some examples are patient associations, relevant databases,

OMIM/proteins/genes links, author's websites, etc...

13) As part of the EMBO Publications transparent editorial process initiative (see our policy here: https://www.embopress.org/transparent-process#Review_Process), EMBO Molecular Medicine will publish online a Peer Review File (PRF) to accompany accepted manuscripts. This file will be published in conjunction with your paper and will include the anonymous referee reports, your point-by-point response and all pertinent correspondence relating to the manuscript. Let us know whether you agree with the publication of the PRF and as here, if you want to remove or not any figures from it prior to publication. Please note that the Authors checklist will be published at the end of the PRF.

14) Please provide a point-by-point letter INCLUDING my comments as well as the reviewer's reports and your detailed responses (as Word file).

I look forward to reading a new revised version of your manuscript as soon as possible.

Yours sincerely,

Poonam Bheda

Poonam Bheda, PhD
Scientific Editor
EMBO Molecular Medicine

***** Reviewer's comments *****

Referee #2 (Comments on Novelty/Model System for Author):

The authors have addressed all my questions sincerely. Readers can justify if the method is practical for clinical application based on the data shown in this pilot study.

Referee #2 (Remarks for Author):

I have no further question.

Referee #3 (Comments on Novelty/Model System for Author):

The technical quality of the manuscript has been improved considerably with the additional data and clearer presentation of results and discussion of limitations.

Referee #3 (Remarks for Author):

I think that the additional data and restructuring of the manuscript, particularly the clearer discussion of the limitations of the approaches has improved the study greatly. This is interesting data which will be of value to the field.

The authors addressed the minor editorial issues.

14th May 2024

Dear Dr. Keller,

Please find enclosed the final reports on your manuscript. We are pleased to inform you that your manuscript is accepted for publication and is now being sent to our publisher to be included in the next available issue of EMBO Molecular Medicine.

Yours sincerely,

Poonam Bheda, PhD
Scientific Editor
EMBO Molecular Medicine
